# SAQ: Stabilizer-Aware Quantum Error Correction Decoder

**David Zenati, Eliya Nachmani**
School of Electrical and Computer Engineering
Ben-Gurion University of the Negev
`znatid@post.bgu.ac.il, eliyanac@bgu.ac.il`

## Abstract

Quantum Error Correction (QEC) decoding faces a fundamental accuracy-efficiency tradeoff. Classical methods like Minimum Weight Perfect Matching (MWPM) exhibit variable performance across noise models and suffer from polynomial complexity, while tensor network decoders achieve high accuracy but at prohibitively high computational cost. Recent neural decoders reduce complexity but lack the accuracy needed to compete with computationally expensive classical methods. We introduce SAQ-Decoder, a unified framework combining transformer-based learning with constraint aware post-processing that achieves both near Maximum Likelihood (ML) accuracy and linear computational scalability with respect to the syndrome size. Our approach combines a dual-stream transformer architecture that processes syndromes and logical information with asymmetric attention patterns, and a novel differentiable logical loss that directly optimizes Logical Error Rates (LER) through smooth approximations over finite fields. SAQ-Decoder achieves high accuracy decoding, with error thresholds of 10.99% (independent noise) and 18.6% (depolarizing noise) on toric codes that closely approach the theoretical ML bounds of 11.0% and 18.9% while outperforming existing neural and classical baselines in accuracy, complexity, and parameter efficiency. Our findings establish that learned decoders can simultaneously achieve competitive decoding accuracy and computational efficiency, addressing key requirements for practical fault-tolerant quantum computing systems.

## 1 Introduction

Since Feynman's 1982 vision of quantum computation Feynman (2018), significant progress has demonstrated that quantum computers can leverage quantum mechanical principles to achieve fundamental computational advantages over classical methods (Steane, 1998; Ladd et al., 2010; Preskill, 2012; deMarti iOlius et al., 2024). Landmark quantum algorithms have demonstrated computational advantages, including exponential speedup for factoring (Shor, 1994) and quadratic search improvement (Grover, 1996). Recent experimental demonstrations of quantum supremacy have further validated quantum computing's potential across diverse domains (Arute et al., 2019; Zhong et al., 2020; Wu et al., 2021; Huang et al., 2022; Madsen et al., 2022; Bao et al., 2023; Bluvstein et al., 2024; Aghaee Rad et al., 2025). These advances promise to revolutionize cryptography (Ekert, 1991; Bennett & Brassard, 2014), optimization (Kadowaki & Nishimori, 1998; Bharti et al., 2022), materials science (Lloyd, 1996), and machine learning (Huang et al., 2022; Cerezo et al., 2022).

Yet, for practical quantum computation to become a reality, errors on the physical level must be corrected with high confidence. Despite recent advances, quantum noise remains a major obstacle to practical quantum computing (goo, 2023). These errors arise through numerous mechanisms: quantum gates cause unwanted errors due to imprecise implementation (Fowler et al., 2012a), while additional errors stem from imperfections in the equipment (Preskill, 2018), interaction with the surrounding environment (Burnett et al., 2019; Etxezarreta Martinez et al., 2021), or measuring quantum systems (Fowler et al., 2012a). While fault-tolerant quantum computation can theoretically be achieved through redundancy by combining multiple physical qubits into one logical qubit (Shor, 1995; Nielsen & Chuang, 2010), this approach creates a critical computational bottleneck: QEC requires real-time decoding algorithms that must process syndrome measurements and determine

corrections within microsecond timescales while maintaining near-optimal accuracy (Terhal, 2015; Higgott, 2022). Current decoding methods face a fundamental trade-off between computational efficiency and error-correction performance, methods like MWPM (Fowler, 2013), Belief Propagation with Ordered Statistics Decoding (BP-OSD) (Roffe et al., 2020) and tensor network decoder (Bravyi et al., 2014) scaling prohibitively with code distance while faster heuristics sacrifice the accuracy essential for fault-tolerant operation (deMarti iOlius et al., 2024). The field of QEC has advanced significantly, with several families of QEC codes proposed, including topological codes (Kitaev, 2003; Bombin & Martin-Delgado, 2006; Fowler et al., 2012a; Chamberland et al., 2020), Quantum Low-Density Parity Check (QLDPC) codes (MacKay et al., 2004; Panteleev & Kalachev, 2021; Breuckmann & Eberhardt, 2021), and quantum turbo codes (Poulin et al., 2009). Recently, there has been significant growth in machine learning techniques applied to quantum decoding (Wang & Tang, 2024; Klusch et al., 2024). However, existing neural decoders typically fail to achieve near-optimal error thresholds, creating a gap between the theoretical potential of learned approaches and the performance requirements of fault-tolerant quantum computing. We address this challenge by introducing a unified framework that combines transformer-based neural decoding with specialized architectural innovations. Our approach leverages neural networks to learn syndrome-to-error mappings while employing dual-stream processing and logical-centric loss design to directly optimize logical error suppression. To achieve this, our framework introduces several key innovations:

- A novel dual-stream transformer architecture (Vaswani et al., 2017) that simultaneously processes syndrome and logical information streams with specialized attention mechanisms, featuring global tokens (Zaheer et al., 2020) and structured masking patterns that capture the geometric constraints and local correlations inherent in stabilizer codes.

- A novel logical-centric multi objective loss, including differentiable minimum entropy loss that directly optimizes LER through smooth approximations of discrete GF(2) constraints, enabling end-to-end training that circumvents the non-differentiability challenges in QEC.

- Constraint-Projected Nullspace Descent (CPND), a novel deterministic post processing algorithm that leverages transformer probabilities as reliability weights to construct recovery operators with exact syndrome consistency.

- Near-optimal error thresholds of 10.99% and 18.6% for toric codes under independent and depolarizing noise, approaching ML bounds 11.0% and 18.9%, with linear scalability in syndrome size and general applicability across stabilizer code families, contrasting favorably with polynomial-scaling classical methods.

Our results significantly outperform existing neural decoders like QEC Transformer (QECCT) (Choukroun & Wolf, 2024) and classical methods like MWPM, while matching the performance of computationally expensive approaches across both toric and rotated surface codes.

The remainder of this paper is organized as follows. Section 2 surveys related work in QEC. Section 3 provides essential background on the quantum decoding problem. Our unified framework is presented in Section 4, where we detail the transformer architecture and dual-stream design and our loss formulation. Section 5 presents comprehensive experimental evaluation. Finally, Section 6 summarizes our contributions and discusses implications for fault-tolerant quantum computing.

## 2 RELATED WORKS

A broad suite of QEC codes has been devised to protect quantum information from decoherence, noise, and gate imperfections. Extracting the underlying logical state from these codes requires dedicated decoders that infer the likely errors from the measured syndromes and prescribe corrections (Dennis et al., 2002). However, ML decoding for quantum codes is NP-hard (Kuo & Lu, 2020), prompting the adoption of approximate methods that trade optimality for computational tractability (deMarti iOlius et al., 2024). Classical quantum decoding approaches include MWPM, which achieves near-optimal thresholds under independent noise but suffers from poor scaling even with practical approximations (Edmonds, 1965; Fowler et al., 2012b; Meinerz et al., 2022); belief propagation, effective for sparse parity-check codes but impeded by quantum degeneracy (Pearl, 2022; Panteleev & Kalachev, 2021; Wang & Tang, 2024); union-find decoders that map syndromes to graph problems but achieve lower thresholds than MWPM (Delfosse & Nickerson, 2021); and tensor-network decoders that attain the highest accuracy at steep computational cost (Bravyi et al.,

2014; goo, 2023). Despite their foundational role, these conventional approaches exhibit inherent limitations that impede practical deployment in large-scale, fault-tolerant quantum systems (Krenn et al., 2023; deMarti iOlius et al., 2024). Machine learning has emerged as a compelling alternative, with various architectures that demonstrate accuracy and speed gains over classical baselines while allowing adaptation to device-specific, correlated noise processes that challenge traditional decoders (Wang & Tang, 2024; deMarti iOlius et al., 2024; Varsamopoulos et al., 2017; 2019; Harper et al., 2020; Magesan & Gambetta, 2020; Liu & Poulin, 2019). Specifically, these architectures are employed in reinforcement learning, (Colomer et al., 2020; Sweke et al., 2020; Fitzek et al., 2020; Çelikkanat et al., 2022; Veeresh et al., 2024), and supervised learning (Bishop & Nasrabadi, 2006; Goodfellow et al., 2016), where models are trained on labeled datasets to map measured syndromes to recovery operations (deMarti iOlius et al., 2024; Wang & Tang, 2024). Early approaches included feedforward networks (Torlai & Melko, 2017), neural decoders learning error distributions (Krastanov & Jiang, 2017), and quantum autoencoders (Locher et al., 2023), demonstrating generalization while reducing complexity and adapting to noise. CNN-based decoders achieve strong performance on topological codes via spatial correlations (Maskara et al., 2019; Meinerz et al., 2022). More recently, transformer-based architectures have been explored, most notably the QECCT (Choukroun & Wolf, 2024), outperforming MWPM across topological codes. While dual-stream transformers like CrossMPT (Park et al., 2024) have been developed for classical codes, they rely on symmetric, locally-constrained attention to emulate belief propagation, lacking the asymmetric global context necessary to address quantum degeneracy. Another innovative AI-based decoder is AlphaQubit (Bausch et al., 2024; Senior et al., 2025), which represents a major milestone in QEC decoding but employs a recurrent structure and processes analog measurement data, unlike our feed-forward architecture which utilizes discrete binary syndrome inputs.

## 3 BACKGROUND

A binary linear code $\mathcal{C} \subseteq GF(2)^n$ is defined as the nullspace of a parity-check matrix $\mathbf{H} \in GF(2)^{(n-k) \times n}$, where $n \in \mathbb{N}$ physical bits encode $k \in \mathbb{N}$ logical (message) bits. For an error vector $\mathbf{e} \in GF(2)^n$, the syndrome $\mathbf{s} = \mathbf{H}\mathbf{e}^\mathsf{T}$ serves as a sufficient statistic for ML decoding. The transition to QEC introduces fundamental complications absent in classical settings. Unlike classical bits that exist in definite states $\{0, 1\}$, quantum information is encoded in qubits—two-level quantum systems that exist in coherent superpositions:

$$|\psi\rangle = \alpha|0\rangle + \beta|1\rangle, \quad \text{where } \alpha, \beta \in \mathbb{C}, \ |\alpha|^2 + |\beta|^2 = 1 \tag{1}$$

This quantum nature creates fundamental challenges: quantum errors form a continuous group, and the no-cloning theorem eliminates classical redundancy. Fortunately, the Pauli channel provides a tractable error model. Any single-qubit error can be decomposed in the Pauli basis $\{I, X, Y, Z\}$:

$$I|\psi\rangle = \alpha|0\rangle + \beta|1\rangle; \ X|\psi\rangle = \alpha|1\rangle + \beta|0\rangle; \ Y|\psi\rangle = -i\alpha|1\rangle + i\beta|0\rangle; \ Z|\psi\rangle = \alpha|0\rangle - \beta|1\rangle \tag{2}$$

A general single-qubit Pauli channel applies error $P \in \{I, X, Y, Z\}$ with probability $\phi_P$, where $\sum_\phi \phi_P = 1$. For $n$ qubits, errors are tensor products $E = P_1 \otimes \cdots \otimes P_n$, leading to $4^n$ possible error patterns. The exponential growth in error patterns ($4^n$ vs. $2^n$ classically) creates a rich combinatorial optimization problem well which suited to neural approaches.

**Stabilizer Formalism.** The stabilizer framework, (Gottesman, 1997), addresses QEC challenges by discretizing the error space while preserving quantum coherence. This formalism exploits the algebraic structure of the Pauli group to construct quantum codes syndrome extraction. The Pauli group foundation. The $n$-qubit Pauli group captures all local quantum errors:

$$\mathcal{P}_n = \big\{ \omega P_1 \otimes \cdots \otimes P_n : \omega \in \{\pm 1, \pm i\}, \ P_j \in \{I, X, Y, Z\} \quad \text{for } j = 1, \ldots, n \big\} \tag{3}$$

The global phases $\omega$ leave syndrome measurements invariant and can be quotiented out.

A stabilizer group $\mathcal{S}$ forms an abelian subgroup of $\mathcal{P}_n$ with $-I \notin \mathcal{S}$. The abelian structure guarantees that all stabilizer elements commute, enabling simultaneous measurability. An $[[n, k, L_{\text{code}}]]$ stabilizer code with distance $L_{\text{code}}$ uses $m = n - k$ independent generators $\{S_i\}_{i=1}^m$ whose joint $+1$ eigenspace defines the codespace:

$$\mathcal{C}_\mathcal{S} = \{|\psi\rangle \in \mathcal{H}_2^n : S_i|\psi\rangle = |\psi\rangle, \quad \text{for } i = 1, \ldots, m\} \tag{4}$$

For an error $E \in \mathcal{P}_n$, the syndrome $s(S_i, E)$ indicates whether stabilizer $S_i$ commutes (0) or anticommutes (1) with $E$. The full syndrome vector $\mathbf{s}(E) = (s(S_1, E), \ldots, s(S_m, E)) \in \{0,1\}^m$ provides a classical signature of the quantum error. Crucially, measuring these stabilizers is non demolition, i.e., extracting error information without disturbing the encoded quantum state. Quantum degeneracy occurs when multiple distinct errors produce identical syndromes because they differ by logical operators that commute with all stabilizers yet act nontrivially on the codespace.

Quantum degeneracy creates a prediction problem: given syndrome $\mathbf{s}$, determine which logical coset contains the true error. This presents formidable computational challenges with exponential syndrome spaces ($2^{O(L_{\text{code}}^2)}$ for surface codes), making neural approaches particularly attractive for learning optimal syndrome-to-coset mappings. Surface codes possess inherent geometric structure ideal for neural learning, with local syndrome correlations and hierarchical error patterns that align perfectly with attention mechanisms capable of capturing both local and global correlations.

## 4 SAQ DECODER

We address QEC problem: given syndrome measurements, predict recovery operations that restore correct logical states. Due to degeneracy, multiple errors yield identical syndromes, requiring decoders that find logically equivalent recovery operations. To tackle this challenge, we propose a novel architecture consists of three sequential stages: (i) dual-stream representation construction, (ii) Syndrome-Logical Transformer Decoder (SLTD), (iii) the post-processing CPND stage and (iv) novel differentiable logical centric loss.

The dual-stream representation construction stage takes syndrome measurements as input and generates initial logical class estimates. These estimates, along with the original syndrome measurements, are then transformed into two token streams that serve as input to the SLTD. Using shared transformer weights, the SLTD processes these streams with distinct attention patterns tailored for QEC: syndrome tokens capture local correlations between neighboring stabilizer measurements, while logical tokens integrate information globally to determine error classes. The dual-stream architecture explicitly models the asymmetric information flow in quantum decoding, from local syndrome violations to global logical error determination.

The SLTD outputs logical class predictions and qubit flip predictions, which are trained using differentiable logical centric losses that approximate discrete GF(2), before being fed to the CPND stage. Subsequently, the CPND enforces syndrome consistency while preserving the transformer's learned representations, ensuring valid QEC.

### 4.1 STAGE 1: DUAL-STREAM REPRESENTATION CONSTRUCTION.

Given a syndrome vector $\mathbf{s} \in \{-1, +1\}^m$, we first obtain an initial logical class estimate $\tilde{\ell} \in \mathbb{R}^{4^k}$ through a shallow MLP $b_\phi : \{-1, +1\}^m \rightarrow \mathbb{R}^{4^k}$, expressed as

$$\tilde{\boldsymbol{\ell}} = b_\phi(\mathbf{s}) \tag{5}$$

where $4^k$ represents the total number of logical equivalence classes for $k$ logical qubits. The shallow MLP $b_\phi(\mathbf{s})$ provides informed priors about the most likely logical class, enabling the SLTD to refine these estimates rather than exploring the entire logical space from scratch. Such a mapping, where a syndrome input is processed by a shallow feed-forward network, appeared in earlier works, notably the FFN layer in (Meinerz et al., 2022) and the initial noise estimator in QECCT (Choukroun & Wolf, 2024). Crucially, in our work, $b_\phi$ performs a global estimation of the logical class ($\tilde{\ell}$), serving as a global prior input to the Logical Stream ($\mathbf{T}_L$). This contrasts with related approaches that utilize these initial layers primarily for generating local physical error probabilities or extracting an initial prediction of the recovery operator. The design choice aligns with the stabilizer formalism, where error correction decisions are made purely based on syndrome information, independent of the protected quantum information. With both syndrome measurements $\mathbf{s}$ and initial logical class estimates $\tilde{\ell}$ available, we construct two complementary token streams that capture different aspects of the quantum decoding problem:

**Syndrome Stream Construction.** Each syndrome measurement $s_i \in \{-1, 1\}$ for $i = 1 \ldots m$ is mapped to a learned embedding $\mathbf{t}_{i,S}^{[0]} = s_i \mathbf{w}_i^S \in \mathbb{R}^d$ using learnable positional embeddings $\mathbf{w}_i^S \in \mathbb{R}^d$

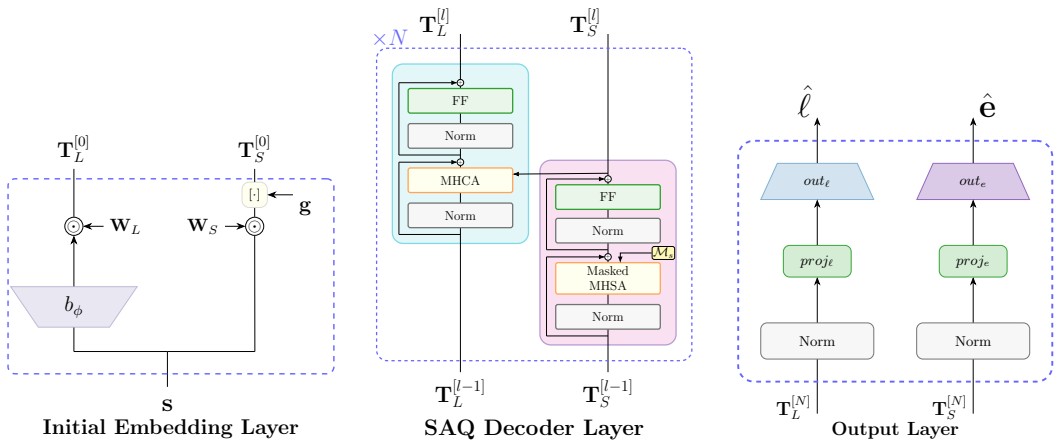

Figure 1: Architecture of SAQ-Decoder.

(where $d$ is the embedding dimension), which collectively form the learnable syndrome embedding matrix $\mathbf{W}_S = [\mathbf{w}_1^S; \ldots; \mathbf{w}_m^S] \in \mathbb{R}^{m \times d}$. A learnable global token $\mathbf{g} \in \mathbb{R}^d$ is then prepended to enable cross-syndrome information exchange, forming the complete syndrome stream: $\mathbf{T}_S^{[0]} = [\mathbf{g}; \mathbf{t}_{1,S}^{[0]}; \ldots; \mathbf{t}_{m,S}^{[0]}] \in \mathbb{R}^{(m+1) \times d}$. The global token enables efficient information aggregation across distant syndrome regions—essential for handling correlated noise and large error clusters.

**Logical Stream Construction.** Predicted logical class logits are embedded as $\mathbf{t}_{j,L}^{[0]} = \tilde{\ell}_j \mathbf{w}_j^L \in \mathbb{R}^d$ for $j = 1 \ldots 4^k$ using learnable class-specific representations $\mathbf{w}_j^L \in \mathbb{R}^d$. These embeddings form the matrix $\mathbf{W}_L = [\mathbf{w}_1^L; \ldots; \mathbf{w}_{4^k}^L] \in \mathbb{R}^{4^k \times d}$ and yield the logical token sequence $\mathbf{T}_L^{[0]} = [\mathbf{t}_{1,L}^{[0]}; \ldots; \mathbf{t}_{4^k,L}^{[0]}] \in \mathbb{R}^{4^k \times d}$. The dual-stream design reflects that syndrome measurements encode local constraint violations while logical estimates capture global degeneracy patterns.

## 4.2 STAGE 2: SYNDROME-LOGICAL TRANSFORMER DECODER (SLTD)

Having constructed dual token streams, $\mathbf{T}_S^{[0]}$ and $\mathbf{T}_L^{[0]}$, the SLTD refines these representations through $N$ transformer layers with asymmetric attention. We detail each computational step for an arbitrary layer $l$. Both token streams first undergo layer normalization (Ba et al., 2016) before attention computation. The normalized tokens are processed using an asymmetric attention mechanism that captures the fundamental information flow in QEC: syndrome measurements reflect local physical constraints, while logical error determination requires global integration. This design restricts syndrome attention to topological neighborhoods while allowing logical tokens global access, enabling efficient local-global information processing. Syndrome self-attention processes syndrome tokens through:

$$\mathbf{Q}_S^{[l-1]} = \widetilde{\mathbf{T}}_S^{[l-1]} \mathbf{W}_Q^{[l]}; \quad \mathbf{K}_S^{[l-1]} = \widetilde{\mathbf{T}}_S^{[l-1]} \mathbf{W}_K^{[l]}; \quad \mathbf{V}_S^{[l-1]} = \widetilde{\mathbf{T}}_S^{[l-1]} \mathbf{W}_V^{[l]} \tag{6}$$

$$\mathbf{A}_S^{[l]} = \text{Softmax}\left( d^{-1/2} \left( \mathbf{Q}_S^{[l-1]} \mathbf{K}_S^{[l-1]^T} + \mathcal{M}_S \right) \right) \mathbf{V}_S^{[l-1]} \tag{7}$$

We introduce a novel syndrome attention mask $\mathcal{M}_S$ that enforces topological constraints:

$$\mathcal{M}_S[i,j] = \begin{cases} 0 & \text{if } (\mathbf{H}\mathbf{H}^T + \mathbf{I}_m)_{i,j} > 0 \text{ or } i = 0 \text{ or } j = 0 \\ -\infty & \text{otherwise} \end{cases} \tag{8}$$

where $\mathbf{H} \in \{0,1\}^{m \times n}$ is the parity-check matrix and $\mathbf{I}_m$ is the identity matrix. The mask permits attention between: (i) each syndrome and itself ($\mathbf{I}_m$), (ii) syndrome pairs that share physical qubits ($\mathbf{H}\mathbf{H}^T > 0$), and (iii) all syndromes with the global aggregation token (corresponding to $i, j = 0$). For the logical stream, logical cross-attention enables logical tokens to attend to the updated syndrome representations.

$$\mathbf{Q}_L^{[l-1]} = \widetilde{\mathbf{T}}_L^{[l-1]} \mathbf{W}_Q^{[l]}; \quad \mathbf{K}_S^{[l]} = \mathbf{T}_S^{[l]} \mathbf{W}_K^{[l]}; \quad \mathbf{V}_S^{[l]} = \mathbf{T}_S^{[l]} \mathbf{W}_V^{[l]} \tag{9}$$

$$\mathbf{A}_L^{[l]} = \text{Softmax}\left(d^{-1/2}\left(\mathbf{Q}_L^{[l-1]}\mathbf{K}_S^{[l]T}\right)\right)\mathbf{V}_S^{[l]} \tag{10}$$

Logical tokens employ unrestricted attention patterns, enabling global syndrome integration. Following attention computation, residual connections combine the attention outputs with input tokens. Both streams pass through standard FFNs with $4\times$ expansion and GELU activation (Hendrycks & Gimpel, 2016). Finally, a second residual connection yields the layer outputs. This process transforms initial token representations into refined syndrome and logical embeddings that capture both local correlations and global quantum code structure.

**Output Generation.** Final token representations are normalized and projected to outputs, where syndrome tokens (excluding the global token) produce physical error predictions $\hat{\mathbf{e}} = \mathbf{W}_{\text{out,S}} \cdot (\widetilde{\mathbf{T}}_{S,\text{no-global}}^{[N]} \cdot \mathbf{w}_{\text{pool,S}})$ and logical tokens generate class $\hat{\boldsymbol{\ell}} = \mathbf{W}_{\text{out,L}} \cdot (\widetilde{\mathbf{T}}_L^{[N]} \cdot \mathbf{w}_{\text{pool,L}})$.

## 4.3 STAGE 3: CONSTRAINT-PROJECTED NULLSPACE DESCENT (CPND)

Neural decoders face a constraint challenge: networks learn correlations but cannot guarantee recovery operators satisfy syndrome consistency over GF(2). CPND bridges this gap through constraint enforcement preserving learned representations. It operates via (i) exact projection ensuring syndrome consistency, and (ii) greedy descent using transformer probabilities to guide optimization toward lower-weight solutions. The raw prediction $\hat{\mathbf{e}}$ (its hard decision, $\mathbf{e}^{\text{pred}}$) is not guaranteed to satisfy the input syndrome constraint. Furthermore, the direct logical class prediction $\hat{\boldsymbol{\ell}}$ provides a slightly superior estimate of the logical class than the class implied by the raw error prediction. The definitive output, $\mathbf{e}(\mathbf{s})$, is therefore produced by the CPND stage, which enforces two critical constraints (i) the syndrome constraint $\mathbf{s} = \mathbf{He}(\mathbf{s})$ and (ii) the target logical class $\hat{\boldsymbol{\ell}} = \mathbf{Le}(\mathbf{s})$, where $\mathbf{L} \in \{0,1\}^{2k \times n}$ encodes the logical operators. This stage uses the transformer outputs $\hat{\mathbf{e}}$ as informative priors. A complete derivation and description of the method is provided in Appendix B.

## 4.4 LOGICAL-CENTRIC LOSS DESIGN

Our training objective combines three loss terms addressing quantum degeneracy by minimizing LER via informed priors, direct classification, and differentiable constraint approximation.

**Informed logical priors loss** trains the auxiliary MLP $b_\phi(\mathbf{s})$ to map syndromes to logical classes:

$$\mathcal{L}_{LP} = \text{CE}(\tilde{\boldsymbol{\ell}}, y_{\text{class}}) \tag{11}$$

where $y_{\text{class}}$ encodes the true logical syndrome as a class index, providing informed priors to guide transformer processing.

**Logical class prediction loss** supervises the transformer's refined logical output:

$$\mathcal{L}_{\text{LC}} = \text{CE}(\hat{\boldsymbol{\ell}}, y_{\text{class}}) \tag{12}$$

ensuring accurate logical classification after cross-attention processing.

**Logical-minimum entropy loss.** A key challenge in neural quantum decoding is enforcing the discrete constraint that recovery operators must preserve logical information. Specifically, we require the true error $\mathbf{e}^{\text{true}}$ and the recovery operator $\mathbf{e}^{\text{pred}}$ satisfy $\mathbf{L}(\mathbf{e}^{\text{true}} \oplus \mathbf{e}^{\text{pred}}) = \mathbf{0}$ over GF(2), where $\mathbf{e}^{\text{pred}}$ is hard decision on the logits $\hat{\mathbf{e}}$ and the residual error $\mathbf{r} = \mathbf{e}^{\text{true}} \oplus \mathbf{e}^{\text{pred}}$ must be a stabilizer for successful error recovery. Our contribution is a differentiable approximation to this discrete constraint. We model the probability that each residual bit is flipped as $\Pr(r_i = 1|e_i^{\text{true}}) = q_i = \sigma((1 - 2e_i^{\text{true}})\hat{e}_i)$ for $i = 1, \ldots, n$ and $\sigma$ is the sigmoid function. For each logical operator $\mathbf{L}_i$, the probability of violating the logical constraint is:

$$\Pr\left(\mathbf{L}_i \cdot \mathbf{r} = 1\right) = \Pr\left(\bigoplus_{j \in \chi_i} \mathrm{L}_{i,j} r_j = 1\right) = \tfrac{1}{2}\left[1 - \prod_{j \in \chi_i}(1 - 2q_j)\right] \tag{13}$$

where $\chi_i$ are the non zero elements set in $\mathbf{L}_i$. Our logical-minimum entropy loss minimizes the expected number of logical violations (full derivation is provided in Appendix A):

$$\mathcal{L}_{\text{Entropy}} = -\frac{1}{2k}\sum_{i=1}^{2k}\log\left(1 - \Pr(\mathbf{L}_i \cdot \mathbf{r} = 1)\right) \tag{14}$$

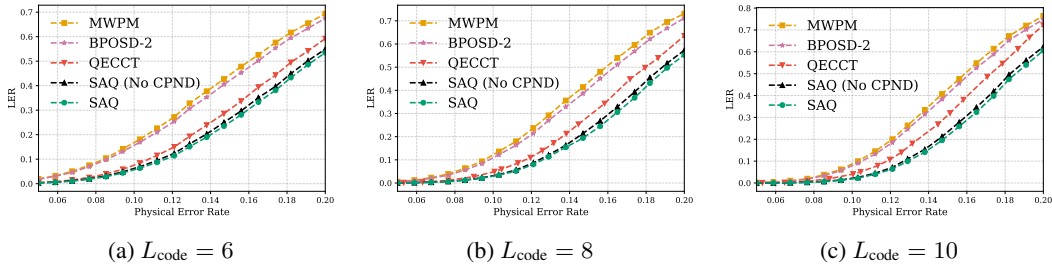

Figure 2: Toric code - depolarizing noise model results.

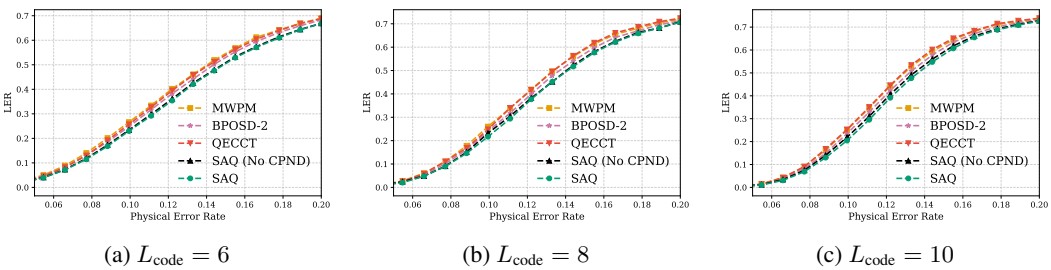

Figure 3: Toric code - independent noise model results.

The combined objective is $\mathcal{L} = \lambda_{\text{LP}}\mathcal{L}_{\text{LP}} + \lambda_{\text{LC}}\mathcal{L}_{\text{LC}} + \lambda_{\text{Entropy}}\mathcal{L}_{\text{Entropy}}$.

## 5 EXPERIMENTS AND RESULTS

To empirically validate the adaptability of the SAQ-Decoder to distinct lattice geometries, we evaluated our method across diverse code families and noise types. We primarily focused on topological codes due to their prominence in fault-tolerant quantum computing (deMarti iOlius et al., 2024), specifically toric codes (Kitaev, 1997) and rotated surface codes (Bombín & Martin-Delgado, 2007). To further demonstrate the framework's broad applicability, we included an evaluation of the Repetition Code (Peres, 1985) and the Color Code (Bombin & Martin-Delgado, 2006) using stim (Gidney, 2021). We evaluated our method under three well-studied noise models: independent noise, depolarizing noise and circuit noise. Detailed descriptions of the code constructions are provided in Appendix C, training details and hyperparameters are provided in Reproducibility Statement. We evaluate our approach against three key baselines: the QECCT (Choukroun & Wolf, 2024), a state-of-the-art neural decoder that outperforms classical methods, Belief Propagation with Order-2 Ordered Statistics Decoder (BPOSD-2) (Roffe et al., 2020), and MWPM algorithm (Fowler, 2013), the gold standard classical decoder for surface codes. The BPOSD family of decoders is widely used, although its worst-case complexity scales as $\mathcal{O}(n^3)$(deMarti iOlius et al., 2024), our implementation leverages the optimized implementation (Roffe, 2022) to provide a strong, practical classical benchmark for quantum codes. Although MWPM has a worst-case complexity $\mathcal{O}(n^3 \log n)$ (deMarti iOlius et al., 2024), we use the optimized implementation from Higgott (2022) which achieves near-quadratic complexity and serves as the primary classical benchmark for topological codes. We also consider the performance of the raw Syndrome Stream prediction, termed SAQ-Decoder (No CPND), as an architectural baseline. As our primary evaluation metric, we use the LER, which measures the probability that the QEC process fails to properly recover the encoded logical information. We also evaluate the code threshold, i.e., the critical noise rate below which increasing code distance improves performance.

### 5.1 EXPERIMENTAL RESULTS

The experiments span code lengths from $L_{\text{code}} = 3$ to 11, comparable to those evaluated in QECCT. These results demonstrate that SAQ-Decoder achieves superior decoding performance compared to state-of-the-art baselines across diverse QEC scenarios. Figure 2 presents the performance of toric

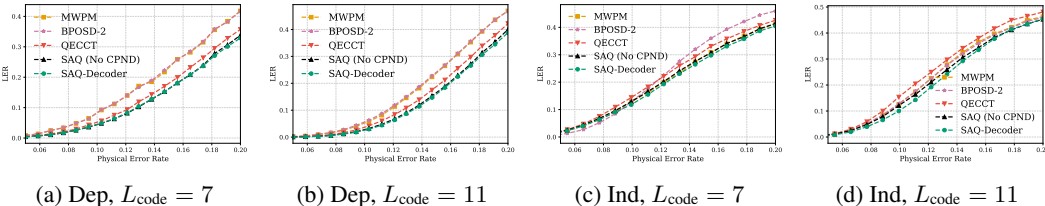

(a) Dep, $L_{\text{code}} = 7$  (b) Dep, $L_{\text{code}} = 11$  (c) Ind, $L_{\text{code}} = 7$  (d) Ind, $L_{\text{code}} = 11$

Figure 4: Rotated surface code results.

codes under depolarizing noise, where SAQ-Decoder exhibits striking advantages. Our method demonstrates consistent superiority across all code distances, with particularly dramatic improvements at $L_{\text{code}} = 10$ where SAQ-Decoder achieves $25 - 50\%$ lower LER compared to MWPM and BPOSD-2 at physical error rates above 0.15. The scalability benefits are clearly evident as the performance gap widens from $L_{\text{code}} = 6$ to $L_{\text{code}} = 10$. Similarly, under independent noise in Figure 3, SAQ-Decoder maintains robust performance across all code sizes, with particularly notable advantages at $L_{\text{code}} = 8$ and $L_{\text{code}} = 10$. Conversely, QECCT exhibits minimal performance gains relative to MWPM and BPOSD-2. Figure 4 demonstrates our method's performance on rotated surface codes under independent and depolarizing noise models. Under depolarization noise in Figures 4a–4b, SAQ-Decoder consistently maintains lower LER than QECCT, BPOSD-2 and MWPM across the entire noise range. Under independent noise conditions in Figures 4c–4d, SAQ-Decoder demonstrates even more substantial improvements, while QECCT exhibits inferior performance compared to MWPM and BPOSD-2. These results validate the effectiveness of our learned decoder with post-processing approach across the spectrum of topological QEC codes and noise models. We attribute SAQ-Decoder's superior performance over QECCT to two key factors: (i) while QECCT focuses on reducing Bit Error Rate (BER), which is not the primary objective in QEC, SAQ-Decoder directly optimizes for logical error suppression; and (ii) our novel architecture combines direct logical class prediction with qubit flip priors, integrating these observations in a post-processing stage that guarantees syndrome consistency—unlike QECCT, which only predicts qubit-level flips without ensuring this crucial constraint. Crucially, the SAQ-Decoder (No CPND) variant consistently outperforms all classical and neural baselines across the entire noise spectrum, demonstrating the power of the dual-stream architecture alone. The SAQ-Decoder achieves error thresholds of 10.99% and 10.7% for toric and rotated surface codes respectively under independent noise, and error thresholds of 18.6% and 18.3% under depolarizing noises. For the toric code under depolarizing noise (Figure 5a), this threshold of 18.6% approaches the ML bound of 18.9% (Bombin et al., 2012) while maintaining linear complexity in syndrome size. This significantly outperforms BPOSD-2 and MWPM (16%) (Wang et al., 2009) and exceeds the previous QECCT result (17.8%). For toric codes under independent noise (Figure 5b), we achieve a threshold of 10.99% while significantly outperforming BPOSD-2 (10.8%) and MWPM (10.3%) (Wang et al., 2003; Higgott, 2022), essentially reaching the ML threshold estimated between 10.9% and 11.0% (ecz, 2024). For rotated surface codes, SAQ-Decoder demonstrates remarkable consistency with the toric code performance. Under depolarizing noise, we achieve a threshold of 18.3% (Figure 5c), significantly exceeding both QECCT (17.2%), BPOSD-2 (14.1%, based on our experiments) and MWPM (14.0%, (deMarti iOlius et al., 2024)), while under independent noise, the threshold reaches 10.7% (Figure 5d) where QECCT achieved 10.3%, BPOSD-2 10.2% and MWPM 10.6%, based on our experiments. These findings indicate that our approach generalizes effectively across different topological code geometries without compromising performance. A detailed comparison of our depolarizing noise threshold against other neural and classical decoders is provided in Appendix E.

To rigorously validate the generalizability of our SAQ-Decoder framework beyond surface codes, we conducted new experiments on two distinct code families: the color code ($L_{code} = 3, 5$) and the repetition code ($L_{code} = 3, 5$). Critically, both experiments were performed under realistic, multi-round circuit-level noise models. These experiments demonstrate robustness confirm our framework's claim of generality, as it is fundamentally agnostic to the code family. Figure 6a illustrates the LER performance on a distance 3 color code with 2 rounds of circuit noise, demonstrating robustness with high marginal gaps from the baselines. In Figure 6b, SAQ-Decoder significantly outperforms all baselines across the entire range of physical error rates. For example, at the highest analyzed rate of $p = 0.02$, SAQ-Decoder achieves a LER that is $17.0\%$ lower than QECCT and

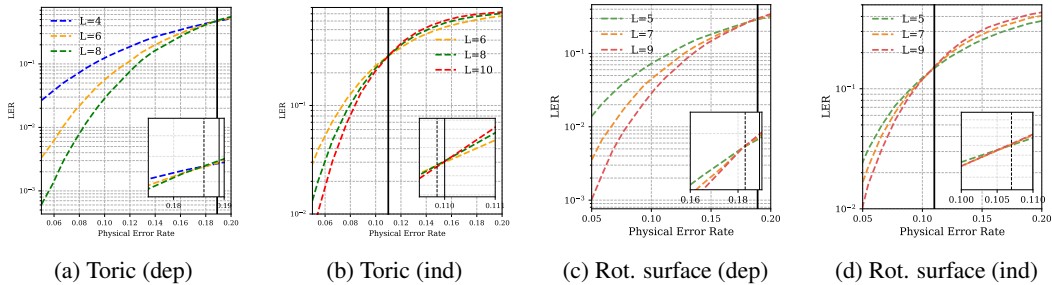

| (a) Toric (dep) | (b) Toric (ind) | (c) Rot. surface (dep) | (d) Rot. surface (ind) |

Figure 5: Error threshold analysis across topological codes and noise models.

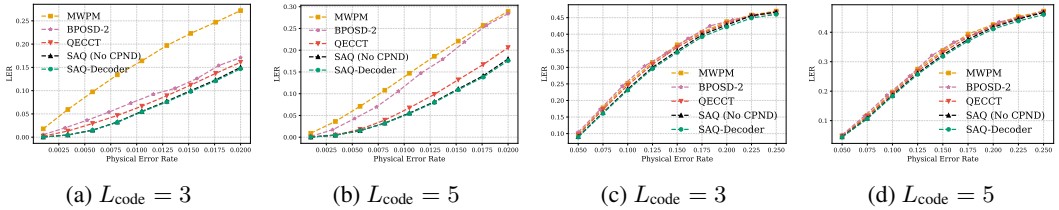

| (a) $L_{\text{code}} = 3$ | (b) $L_{\text{code}} = 5$ | (c) $L_{\text{code}} = 3$ | (d) $L_{\text{code}} = 5$ |

Figure 6: Color code and repetition code with circuit noise results. (a)–(b) are color code results and (c)–(d) are repetition code results.

64.2% lower than the MWPM baseline. Figure 6c presents the LER results for a distance 3 repetition code with 3 rounds of circuit noise, showing that the decoder remains robust with performance gaps relative to the baseline methods. Similarly, Figure 6d shows the results for a distance 5 repetition code with 3 rounds of circuit noise. While the performance of all decoders is closer on this code, SAQ-Decoder consistently maintains the lowest logical error rate. At $p = 0.25$, SAQ-Decoder achieves a LER that is 1.83% lower than QECCT and 2.61% lower than MWPM.

## 5.2 ABLATION STUDIES AND ANALYSIS

To understand the contribution of individual architectural components in our framework, we conduct comprehensive ablation studies on a toric code with distance $L_{\text{code}} = 6$ under depolarizing noise.

**Dual-Stream Study.** To validate our dual-stream design, we conducted an ablation study examining four architectural variants: (1) separate weights per stream within a layer instead of weight sharing across the decoder stack, (2) symmetric cross-attention where both streams attend to each other rather than our asymmetric design, (3) logical-stream-only architecture removing syndrome processing, and (4) syndrome-stream-only architecture removing logical processing, as shown in Figure 7a. The results reveal a clear performance hierarchy from worst to best: logical-stream-only, bidirectional cross-attention, syndrome-stream-only, no weight sharing, and our full SAQ-Decoder architecture. These ablations demonstrate that single-stream variants perform poorly, weight sharing across layers slightly improves efficiency while halving the parameter count, and asymmetric information flow from syndromes to logical inference outperforms symmetric attention, validating the importance of our specialized dual-stream processing design for effective QEC.

**Multi-Loss Ablation.** To investigate the relative importance of different training objectives in our framework, we conduct an ablation study on the loss function components, varying the weighting parameters $\lambda_{LP}$, $\lambda_{LC}$, and $\lambda_{Entropy}$ for logical prior, logical classification, and entropy regularization respectively, as shown in Figure 7b. The full multi-objective formulation $(\lambda_{LP}, \lambda_{LC}, \lambda_{Entropy} = 0.2, 1.0, 1.0)$ achieves the lowest final average LER of 1.972e-01. Systematic removal of individual components reveals measurable performance degradation: removing logical classification increases LER to 2.113e-01 (+7.2%), removing logical prior to 2.055e-01 (+4.2%), and removing entropy regularization to 2.047e-01 (+3.8%). These results confirm that all three objectives contribute meaningfully to the model's QEC performance, with logical classification being the most critical component.

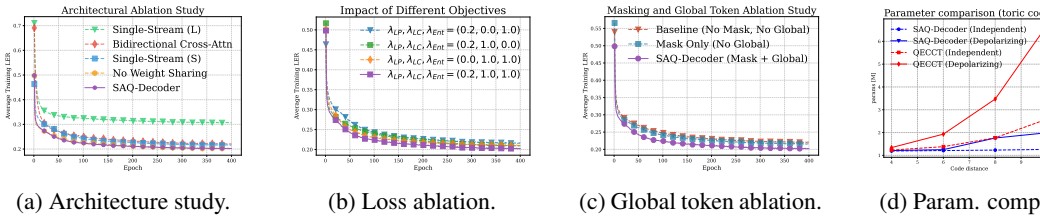

|   |   |   |   |
|---|---|---|---|
| (a) Architecture study. | (b) Loss ablation. | (c) Global token ablation. | (d) Param. comp. |

Figure 7: Ablation studies results.

**Effect of Global Token.** The inclusion of a global token (*SAQ-Decoder*) improves both training dynamics and final performance compared to the masked architecture without a global token (*Mask Only*), as shown in Figure 7c. The full SAQ-Decoder achieves faster convergence and lower final LER ($\sim 0.19$ versus $\sim 0.21$ average LER). Notably, attention masking itself provides substantial benefits, with the mask-only architecture significantly outperforming the unmasked baseline (*Baseline*, neither mask nor global token). The global token acts as a syndrome-level aggregator, enabling the model to capture global syndrome patterns that local interactions might miss.

**Computational Complexity.** Our model achieves $O(Nmd^2)$ time complexity per forward pass by exploiting sparse attention patterns, avoiding the naive $O(Nm^2d)$ complexity of dense attention. The subsequent CPND refinement requires $O(m)$ time for online inference. This yields optimal linear scaling in the syndrome length and quadratic scaling in code distance. The $2^{2k}$ embedding term is negligible since $k$ is small in practice (e.g., $k = 1$ for surface codes and $k = 2$ for toric codes). Our numerical comparison in Table 1 demonstrates that the SAQ-Decoder achieves significantly lower FLOPs and faster inference time compared to QECCT, validating its suitability for real-time QEC decoding. Extended data is available in Appendix F.

Table 1: Abridged Numerical Complexity Comparison (Toric Code)

| Metric | L=6 (Depol) | | L=10 (Depol) | |
|---|---|---|---|---|
| | **SAQ-Decoder** | **QECCT** | **SAQ-Decoder** | **QECCT** |
| Total FLOPs [G ↓] | **0.21** | 1.05 | **0.80** | 4.10 |
| Inference Time [ms ↓] | **1.2** | 7.0 | **4.5** | 20.1 |

**Parameter Efficiency.** Figure 7d demonstrates that SAQ-Decoder for toric code maintains near-constant parameter count ($1.2 - 1.9M$) across code distances $L_{\text{code}} = 4$ to $L_{\text{code}} = 10$ for both noise models, exhibiting excellent scalability. In contrast, QECCT suffers from significant parameter growth, reaching $6.71M$ parameters at $L_{\text{code}} = 10$ under depolarizing noise, which is a $3.5\times$ increase over our method. This difference stems from fundamental architectural choices: while QECCT processes both qubit and syndrome information through transformer layers, our approach leverages logical class prediction to decouple syndrome processing from the full qubit space dimensionality. Although both methods employ sparse attention patterns, SAQ-Decoder's logical embedding strategy avoids the quadratic scaling in the qubit-syndrome space that affects QECCT.

## 6 CONCLUSION

We introduced SAQ-Decoder, a unified QEC framework that combines learned syndrome-to-error mappings with exact syndrome constraint satisfaction. Our dual-stream architecture with asymmetric attention captures the geometric structure of stabilizer codes while maintaining linear computational complexity in syndrome size. Experimental results demonstrate error thresholds of 10.99% and 18.6% for toric codes under independent and depolarizing noise respectively, essentially approaching ML bounds while significantly outperforming existing neural and classical decoders. The framework's parameter efficiency and general applicability to any stabilizer code family make it a practical solution for scaling fault-tolerant quantum computation, bridging the critical gap between neural pattern recognition and the structured optimization requirements of QEC.

REPRODUCIBILITY STATEMENT

**Logical-minimum entropy loss.** Due to page limitations, we provide the full derivation and exposition of the logical-minimum entropy loss in Appendix A.

**CPND stage.** A detailed derivation and exposition of the CPND stage is presented in Appendix B.

**Training Details.** For reproducibility, we detail our complete training methodology and hyperparameters in Appendix D, with full source code provided in the Supplementary Materials. Our source code is publicly available on GitHub at: `https://github.com/DavidZenati/SAQ-Decoder/tree/main`.

ACKNOWLEDGEMENT

The contribution of David Zenati is part of a Ph.D. thesis research conducted at Ben Gurion University of the Negev.

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

## A  LOGICAL-MINIMUM ENTROPY LOSS DERIVATION

The goal of decoding in stabilizer codes is to produce a correction $\mathbf{e}^{\mathrm{pred}}$ (hard decision on the logits $\hat{\mathbf{e}}$) such that the combined error $\mathbf{r} = \mathbf{e}^{\mathrm{true}} \oplus \mathbf{e}^{\mathrm{pred}}$ is a stabilizer, hence acts trivially on all logical qubits. Equivalently, with $\mathbf{L}$ denoting the logical-operator matrix, the logical-coset constraint is

$$\mathbf{L}\left(\mathbf{e}^{\mathrm{true}} \oplus \mathbf{e}^{\mathrm{pred}}\right) = \mathbf{0} \text{ over } GF(2) \tag{15}$$

so decoding succeeds if no logical parity flips. Optimizing this condition directly is difficult because XOR is non-differentiable. We therefore derive a smooth, probability-calibrated surrogate that replaces hard parities by differentiable sign-expectations of Bernoulli logits. The resulting logical minimum-entropy loss maximizes the probability that each logical parity is zero while preserving the exact coset semantics in expectation and providing stable, well-aligned gradients for end-to-end training.

For each qubit $i$, let $\hat{e}_i \in \mathbb{R}$ be the model logit and

$$p_i \triangleq \sigma(\hat{e}_i) = \frac{1}{1 + e^{-\hat{e}_i}} \tag{16}$$

the corresponding flip probability, where $\sigma(\cdot)$ is the sigmoide function. We model the predicted error bit as

$$e_i^{\mathrm{pred}} \sim \mathrm{Bernoulli}(p_i), \tag{17}$$

while the ground-truth bit $e_i^{\mathrm{true}} \in \{0, 1\}$ is fixed for the given sample. Define the per-qubit XOR

$$r_i \triangleq e_i^{\mathrm{true}} \oplus e_i^{\mathrm{pred}} \in \{0, 1\}. \tag{18}$$

Conditioning $r_i$ on $e_i^{\mathrm{true}}$, we have

$$r_i | e_i^{\mathrm{true}} \sim \begin{cases} Bernoulli\left(\sigma(\hat{e}_i)\right) & \text{if } e_i^{\mathrm{true}} = 0 \\ Bernoulli\left(1 - \sigma(\hat{e}_i)\right) & \text{if } e_i^{\mathrm{true}} = 1 \end{cases} \tag{19}$$

it holds since for $r_i = 0 \oplus e_i^{\mathrm{pred}} = e_i^{\mathrm{pred}}$ and $r_i = 1 \oplus e_i^{\mathrm{pred}} = 1 - e_i^{\mathrm{pred}}$.

We focus on the non-parity conditional error probability

$$q_i \triangleq Pr(r_i = 1 | e_i^{\mathrm{true}}) = (1 - e_i^{\mathrm{true}})\sigma(\hat{e}_i) + (1 - \sigma(\hat{e}_i))e_i^{\mathrm{true}} \tag{20}$$

which our loss is designed to minimize; driving $q_i \to 0$ forces the prediction to agree with the ground truth modulo stabilizers (i.e., no logical flip).

We now simplify $q_i$, starting from

$$q_i = \Pr(r_i = 1 | e_i^{\mathrm{true}}) = \frac{1 - e_i^{\mathrm{true}}}{1 + \exp(-\hat{e}_i)} + \frac{\exp(-\hat{e}_i) \cdot e_i^{\mathrm{true}}}{1 + \exp(-\hat{e}_i)} \tag{21}$$

$$= \frac{1 - e_i^{\mathrm{true}} + \exp(-\hat{e}_i) \cdot e_i^{\mathrm{true}}}{1 + \exp(-\hat{e}_i)} \tag{22}$$

Let $a = 1 - 2e_i^{\mathrm{true}} \in \{-1, 1\}$, so $e_i^{\mathrm{true}} = (1 - a)/2$, focusing on the numerator

$$1 - e_i^{\mathrm{true}} + \exp(-\hat{e}_i) \cdot e_i^{\mathrm{true}} = 1 + e_i^{\mathrm{true}} \cdot (\exp(-\hat{e}_i) - 1) \tag{23}$$

$$= 1 + \frac{1 - a}{2} \cdot (\exp(-\hat{e}_i) - 1) \tag{24}$$

$$= 1 + \frac{1}{2} \cdot (\exp(-\hat{e}_i) - 1) - \frac{a}{2} \cdot (\exp(-\hat{e}_i) - 1) \tag{25}$$

$$= \frac{(1 + \exp(-\hat{e}_i)) - a \cdot (\exp(-\hat{e}_i) - 1)}{2} \tag{26}$$

Plugging this back to equation 21 gives

$$q_i = \Pr(r_i = 1 | e_i^{\mathrm{true}}) = \frac{(1 + \exp(-\hat{e}_i)) - a \cdot (\exp(-\hat{e}_i) - 1)}{2 \cdot (1 + \exp(-\hat{e}_i))} \tag{27}$$

$$= \frac{1}{2}\left[1 - a \cdot \frac{\exp(-\hat{e}_i) - 1}{\exp(-\hat{e}_i) + 1}\right] \tag{28}$$

Using $\frac{\exp(-x)-1}{\exp(-x)+1} = -\tanh(x/2)$ with $x = \hat{e}_i$

$$q_i = \frac{1}{2}\left[1 + a \cdot \tanh\left(\frac{\hat{e}_i}{2}\right)\right] \tag{29}$$

Here $\tanh(\cdot)$ is the hyperbolic tangent.

From the previous result,

$$\Pr(r_i = 1 \mid e_i^{\text{true}}) = \frac{1}{2}\left[1 + a \tanh(\hat{e}_i/2)\right], \qquad a = 1 - 2e_i^{\text{true}} \in \{\pm 1\}. \tag{30}$$

Substituting $e_i^{\text{true}} = 1$ (so $a = -1$) gives

$$\Pr(r_i = 1 \mid e_i^{\text{true}} = 1) = \frac{1}{2}\left[1 - \tanh(\hat{e}_i/2)\right] \tag{31}$$

while $e_i^{\text{true}} = 0$ (so $a = +1$) gives

$$\Pr(r_i = 1 \mid e_i^{\text{true}} = 0) = \frac{1}{2}\left[1 + \tanh(\hat{e}_i/2)\right] \tag{32}$$

Since $\tanh$ is odd, i.e., $a \tanh(x) = \tanh(a\,x)$ for $a \in \{\pm 1\}$,

$$\Pr(r_i = 1 \mid e_i^{\text{true}}) = \frac{1}{2}\left[1 + \tanh(a\,\hat{e}_i/2)\right]. \tag{33}$$

To convert the tanh form back to a probability, we use the sigmoid–tanh identity (as shown in Proposition A.1).

**proposition A.1** (Sigmoid–tanh identity). *For all $y \in \mathbb{R}$,*

$$\sigma(y) = \frac{1}{1 + \exp(-y)} = \frac{1}{2}\left[1 + \tanh(y/2)\right] \tag{34}$$

*Proof.* Start from sigmoid $\sigma(y)$, multiplying numerator and denominator by $\exp(y/2)$:

$$\sigma(y) = \frac{1}{1 + \exp(-y)} = \frac{\exp(y/2)}{\exp(y/2) + \exp(-y/2)} \tag{35}$$

Let $A = \exp(y/2)$ and $B = \exp(-y/2)$. Then:

$$\frac{A}{A+B} = \frac{1}{2}\frac{(A+B) + (A-B)}{A+B} = \frac{1}{2}\left(1 + \frac{A-B}{A+B}\right) \tag{36}$$

But

$$\frac{A-B}{A+B} = \frac{\exp(y/2) - \exp(-y/2))}{\exp(y/2) + \exp(-y/2)} = \tanh\left(\frac{y}{2}\right) \tag{37}$$

Hence $\sigma(y) = \frac{1}{2}\left[1 + \tanh(\frac{y}{2})\right]$ $\qquad\square$

Therefore,

$$q_i = \Pr(r_i = 1 \mid e_i^{\text{true}}) = \frac{1}{2}\left[1 + \tanh(a\,\hat{e}_i/2)\right] = \sigma(a\,\hat{e}_i) = \sigma\big((1 - 2e_i^{\text{true}})\hat{e}_i\big) \tag{38}$$

We now pass to logical parities: each row of the logical operator matrix $\mathbf{L}$ (which correspond to a distinct logical operator) induces a parity check over $\mathbf{r}$;

$$\mathbf{L}_i \cdot \mathbf{r} = \bigoplus_{j \in \chi_i} \mathbf{L}_{i,j} r_j \tag{39}$$

where $\chi_i$ are the non zero elements set in $\mathbf{L}_i$, the following proposition provides $\mathbf{L}_i \cdot \mathbf{r}$ distribution in closed form.

**proposition A.2** (Bernoulli parity distribution). *Let $\{X_j\}_{j=1}^n$ be independent Bernoulli random variables with $\Pr(X_j = 1) = q_j \in [0,1]$, and define the GF(2) parity $Q = \bigoplus_{j=1}^n X_j \in \{0,1\}$. Then*

$$\Pr(Q = 1) = \tfrac{1}{2}\Big[1 - \prod_{j=1}^n (1 - 2q_j)\Big] \tag{40}$$

$$\Pr(Q = 0) = \tfrac{1}{2}\Big[1 + \prod_{j=1}^n (1 - 2q_j)\Big] \tag{41}$$

*Equivalently, $\mathbb{E}[(-1)^Q] = \prod_{j=1}^n (1 - 2q_j)$.*

*Proof.* Consider the following product identity

$$(-1)^Q = (-1)^{X_1 \oplus X_2 \oplus \cdots \oplus X_n} = \prod_i (-1)^{X_i} \tag{42}$$

This identity holds because: $(-1)^{0 \oplus 0} = 1 = (-1)^0 \cdot (-1)^0$, $1(-1)^{0 \oplus 1} = -1 = (-1)^0 \cdot (-1)^1$ and $1(-1)^{1 \oplus 1} = 1 = (-1)^1 \cdot (-1)^1$

Taking expectations on both sides yields

$$\mathbb{E}[(-1)^Q] = \mathbb{E}\Big[\prod_i (-1)^{X_i}\Big] \tag{43}$$

and, by expanding the definition of expectation with respect to $Q$,

$$\mathbb{E}[(-1)^Q] = (-1)^0 \cdot \Pr(Q = 0) + (-1)^1 \cdot \Pr(Q = 1) = 1 - 2Pr(Q = 1) \tag{44}$$

Under the independence assumption,

$$\mathbb{E}\Big[\prod_i (-1)^{X_i}\Big] = \prod_i \mathbb{E}[(-1)^{X_i}] \tag{45}$$

and each factor evaluates to

$$\mathbb{E}[(-1)^{X_i}] = (-1)^0 \cdot \Pr(X_i = 0) + (-1)^1 \cdot \Pr(X_i = 1) = (1 - q_i) - q_i = 1 - 2q_i \tag{46}$$

Thus,

$$\mathbb{E}[(-1)^Q] = \prod_i (1 - 2q_i) \tag{47}$$

Equating equation 44 and equation 47 and solving for $\Pr(Q = 1)$ gives

$$\Pr(Q = 1) = \tfrac{1}{2}\Big[1 - \prod_i (1 - 2q_i)\Big] \tag{48}$$

The key insight is that the XOR operation in GF(2) corresponds to multiplication in the group $(\{-1, 1\}, \cdot)$, which enables a closed-form expression for the parity distribution under independence. $\square$

Therefore, it follows from Proposition A.2 and from equation 38 that

$$\Pr(\mathbf{L}_i \cdot \mathbf{r} = 1) = \tfrac{1}{2}\Big[1 - \prod_{j \in \chi_i} (1 - 2q_j)\Big] = \tfrac{1}{2}\Big[1 - \prod_{j \in \chi_i} \big(1 - 2\sigma((1 - 2e_j^{\text{true}})\hat{e}_j)\big)\Big] \tag{49}$$

Since $\mathbf{L}_i \cdot \mathbf{r} = 1$ corresponds to a logical error detected by the $i$-th logical operator (i.e., a bit flip or phase flip on a logical qubit), successful QEC requires minimizing this probability across all logical operators. To achieve this objective, we employ a minimum entropy loss that directly optimizes the negative log-likelihood of the desired outcome, namely that no logical errors occur. This approach

concentrates the probability mass on the correct logical state while heavily penalizing configurations that lead to logical failures.

Substituting the expression for $\Pr(\mathbf{L}_i \cdot \mathbf{r} = 1)$ into the entropy objective yields

$$L_{\text{entropy}} = -\frac{1}{2k} \sum_{i=1}^{2k} \log\Big(1 - \Pr(\mathbf{L}_i \cdot \mathbf{r} = 1)\Big) \tag{50}$$

$$= -\frac{1}{2k} \sum_{i=1}^{2k} \log\Big(1 + \prod_{j \in \chi_i} \big(1 - 2\,\sigma((1 - 2e_j^{\text{true}})\hat{e}_j)\big)\Big) \tag{51}$$

## B  CONSTRAINT-PROJECTED NULLSPACE DESCENT (CPND)

Neural decoders face a constraint challenge: networks learn correlations but cannot guarantee recovery operators satisfy syndrome consistency over GF(2). CPND bridges this gap through constraint enforcement preserving learned representations. It operates via (i) exact projection ensuring syndrome consistency, and (ii) greedy descent using transformer probabilities to guide optimization toward lower-weight solutions.

The augmented matrix $\widehat{\mathbf{H}} = \big[\mathbf{H}; \mathbf{L}\big] \in \{0,1\}^{(m+2k)\times n}$ is constructed by vertically stacking the parity-check matrix $\mathbf{H} \in \{0,1\}^{m\times n}$ (whose rows represent stabilizer generators that, when measured, produce the syndrome) above the logical operator matrix $\mathbf{L} \in \{0,1\}^{2k\times n}$ (whose rows encode logical operators that, when applied to the error vector, determine the logical error class, $k$ X-type $+ k$ Z-type logical operators). Since the stabilizer generators are linearly independent over GF(2) and the logical operators lie in the normalizer but not in the stabilizer subgroup, the rows of $\widehat{\mathbf{H}}$ are linearly independent. Therefore, $\text{rank}(\widehat{\mathbf{H}}) = m + 2k$.

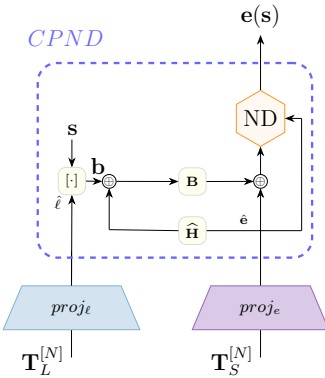

Figure 8: CPND.

The constraint vector is $\mathbf{b} = [\mathbf{s}; \ell] \in \{0,1\}^{(m+2k)}$ where $\mathbf{s}$ is the syndrome and $\ell$ is the binarized logical class prediction. We define the feasible recovery operator $\mathbf{e}(\mathbf{s})$ set

$$\mathcal{F} = \Big\{\mathbf{e}(\mathbf{s}) \in \{0,1\}^n \mid \widehat{\mathbf{H}}\mathbf{e}(\mathbf{s}) = \mathbf{b}\Big\} \tag{52}$$

We precompute a left inverse $\mathbf{B} \in \{0,1\}^{n\times(m+2k)}$ with $\widehat{\mathbf{H}}\mathbf{B} = \mathbf{I}_{m+2k}$, a proof of existence of such $\mathbf{B}$ is given in Proposition B.1.

**proposition B.1** (Existence and constructive computation of a left inverse over GF(2)). *Let $A \in \{0,1\}^{r\times n}$ with $r \leq n$. There exists $X \in \{0,1\}^{n\times r}$ such that $AX = I_r$ if and only if $\text{rank}(A) = r$. Moreover, when $A$ has full row rank, the $r$ column-wise systems*

$$A x_i = e_i, \qquad i = 1, \ldots, r,$$

*(where $e_i$ is the $i$-th standard basis vector of $GF(2)^r$) are all consistent, and any selection of solutions $\{x_i\}_{i=1}^r$ stacked as $X = [x_1 \; \cdots \; x_r]$ satisfies $AX = I_r$.*

*Proof.* ($\Rightarrow$) If $AX = I_r$, then $\text{rank}(A) \geq \text{rank}(I_r) = r$, hence $\text{rank}(A) = r$. ($\Leftarrow$) If $\text{rank}(A) = r$, then the column space $\text{im}(A) \subseteq GF(2)^r$ has dimension $r$ and therefore equals $GF(2)^r$. Thus each $e_i$ lies in $\text{im}(A)$, so there exists $x_i$ with $Ax_i = e_i$. Stacking these solutions yields $AX = [Ax_1 \cdots Ax_r] = [e_1 \cdots e_r] = I_r$. $\square$

Given $\mathbf{e}^{\text{pred}}$, we compute the residual $\mathbf{y} = \mathbf{b} \oplus \widehat{\mathbf{H}}\mathbf{e}^{\text{pred}}$ and apply the projection $\mathbf{e}' = \mathbf{e}^{\text{pred}} \oplus \mathbf{B}\mathbf{y}$. By construction: $\widehat{\mathbf{H}}\mathbf{e}' = \widehat{\mathbf{H}}\mathbf{e}^{\text{pred}} \oplus \mathbf{y} = \mathbf{b}$, ensuring $\mathbf{e}' \in \mathcal{F}$. The projected solution $\mathbf{e}'$ satisfies all constraints but is suboptimal in sense of minimum weight recovery operation, the left inverse $\mathbf{B}$ is constructed purely algebraically and ignores the transformer's learned qubit flip probability. Since the constraint set forms an affine space $\mathbf{e}' \oplus \ker(\widehat{\mathbf{H}})$, we traverse this space to find lower-cost solutions while preserving feasibility. Let $\mathbf{N} = [\mathbf{v}_1, \ldots, \mathbf{v}_g] \in \{0, 1\}^{n \times g}$ span $\ker(\widehat{\mathbf{H}})$ with $g = n - (m + 2k)$. We extract qubit flip probability from transformer predictions: $p_q = \sigma(\hat{\mathbf{e}}_q)$ and define weights as log-likelihood ratios $w_q = -\log(p_q/(1 - p_q))$. The objective is to minimize weighted Hamming cost: $\text{wt}_w(\mathbf{e}) = \sum_{q=1}^{n} w_q e_q$ where $\mathbf{e} \in \{0, 1\}^n$.

Having $\mathbf{e}'$, we want to descend in the nullspace to find a minimal weight solution. First we convert the binary solution $\mathbf{e}'$ to signs $\sigma' = (1 - 2\mathbf{e}') \in \{+1, -1\}^n$, where $\sigma'_q = +1$ if $e'_q = 0$ and $\sigma'_q = -1$ if $e'_q = 1$ for $q = 1 \ldots, n$. The main loop performs a single pass over the $g$ nullspace generators $\{\mathbf{v}_j\}_{j=1}^g$. For each generator $\mathbf{v}_j$, we identify its support $\chi_j = \{q : \mathbf{v}_{j,q} = 1\}$ and compute the cost change $\Delta_j = \sum_{q \in \chi_j} w_q \sigma'_q$. If $\Delta_j < 0$, we accept the move: update $\mathbf{e}' \leftarrow \mathbf{e}' \oplus \mathbf{v}_j$ and flip signs $\sigma'_q \leftarrow -\sigma'_q$ for all $q \in \chi_j$.

Since $\widehat{\mathbf{H}}\mathbf{v}_j = \mathbf{0}$ for $j = 1 \ldots g$, the nullspace descent preserves constraint satisfaction while achieving monotonic cost reduction, terminating at a locally optimal solution. Algorithm 1 presents the complete method.

---

**Algorithm 1** Constraint-Projected Nullspace Descent (CPND)

---

**Require:** $\widehat{\mathbf{H}}, \mathbf{B}, \mathbf{N}$ ; $\mathbf{b}$; $\mathbf{e}^{\text{pred}}$; weights $w \in \mathbb{R}^n$
**Ensure:** $\mathbf{e}(\mathbf{s}) \in \{0, 1\}^n$ with $\widehat{\mathbf{H}}\mathbf{e}(\mathbf{s}) = \mathbf{b}$, and reduced weighted cost $\text{wt}_w(\mathbf{e}(\mathbf{s})) = \sum_q w_q e_q(\mathbf{s})$

1: $\mathbf{y} \leftarrow \mathbf{b} \oplus \widehat{\mathbf{H}}\mathbf{e}^{\text{pred}}$
2: $\mathbf{e}' \leftarrow \mathbf{e}^{\text{pred}} \oplus \mathbf{B}\mathbf{y}$
3: $\sigma' \leftarrow (1 - 2\mathbf{e}') \in \{+1, -1\}^n$
4: **for** $j = 1$ to $g$ **do**
5: $\quad \chi_j \leftarrow \{q : v_{j,q} = 1\}$
6: $\quad \Delta_j \leftarrow \sum_{q \in \chi_j} w_q \sigma'_q$
7: $\quad$ **if** $\Delta_j < 0$ **then**
8: $\quad\quad \mathbf{e}' \leftarrow \mathbf{e}' \oplus \mathbf{v}_j$
9: $\quad\quad \sigma'_q \leftarrow -\sigma'_q$ for all $q \in \chi_j$
10: $\quad$ **end if**
11: **end for**
12: **return** $\mathbf{e}(\mathbf{s}) \leftarrow \mathbf{e}'$

---

Since stabilizer generators commute, yielding $\mathbf{H}\mathbf{H}^{\mathbf{T}} = \mathbf{0}$ over GF(2), the columns of $\mathbf{H}^{\mathbf{T}}$ span a subspace of $\ker(\mathbf{H})$, providing an approximated basis for nullspace descent.

Figure 9 presents decoder recovery operator weights as a function of physical error rate $p \in \{0.05, 0.10, 0.15, 0.20\}$ for the toric code under independent noise ($L_{\text{code}} = 4$). We compare three post-processing approaches that operate on SAQ-Decoder representations: the projection baseline ($\mathbf{e}' = \mathbf{e}^{\text{pred}} \oplus \mathbf{B}\mathbf{y}$), CPND and OSD-0 (Fossorier & Lin, 1995; Roffe et al., 2020). OSD-0 is a post-processing algorithm that achieves minimum weight solutions while maintaining syndrome consistency, but requires matrix inversion operations that scale cubically in complexity, compared to our method's linear complexity. Across all error rates, the methods exhibit consistent performance ordering: OSD-0 achieves the lowest weights, CPND performs comparably to OSD-0, while the projection baseline consistently yields the highest weights. The performance gaps increase monotonically

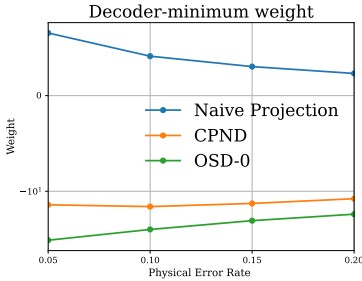

Figure 9: Comparison of recovery operator weights.

with $p$, demonstrating that CPND consistently approaches minimum-weight solutions. These results highlight that structure-aware post-processing methods (CPND and OSD-0) achieve uniformly superior weight minimization compared to naive projection approaches.

## C  SURFACE CODES

We evaluate our method on surface codes due to their prominence in fault-tolerant quantum computing, specifically toric codes (Kitaev, 1997) which encode $k = 2$ logical qubits in $n = 2L_{\text{code}}^2$ physical qubits, and rotated surface codes (Bombín & Martin-Delgado, 2007) which encode $k = 1$ logical qubit in $n = L_{\text{code}}^2$ physical qubits. Toric codes utilize periodic boundary conditions with qubits on lattice edges, while rotated surface codes employ a lattice geometry with qubits on vertices. The stabilizer generators are organized into two distinct groups based on lattice geometry, with different implementations for each code family. For toric codes, vertex stabilizers are constructed as products of Pauli-X operators acting on all qubits adjacent to each lattice vertex, while plaquette stabilizers consist of products of Pauli-Z operators acting on qubits surrounding each lattice face, yielding a total of $m = 2L_{\text{code}}^2 - 2$ stabilizer generators ($L_{\text{code}}^2 - 1$ vertex stabilizers and $L_{\text{code}}^2 - 1$ plaquette stabilizers corresponding to the vertices and faces of the $L_{\text{code}} \times L_{\text{code}}$ toric lattice). Rotated surface codes employ a fundamentally different geometry where all stabilizer generators are placed on lattice faces rather than being split between vertices and plaquettes. These face-based stabilizers come in two alternating types: X-type stabilizers (tensor products of Pauli-X operators on qubits surrounding a face) and Z-type stabilizers (tensor products of Pauli-Z operators on qubits surrounding a face). The lattice arrangement creates a natural checkerboard pattern where X- and Z-type stabilizers alternate, ensuring that every physical qubit—positioned on a vertex of the rotated lattice—participates in exactly two X-type and two Z-type stabilizer measurements. For an $L_{\text{code}} \times L_{\text{code}}$ rotated surface code, this checkerboard arrangement yields $m = 2L_{\text{code}}^2 - 1$ independent stabilizer generators, with $\frac{L_{\text{code}}^2 - 1}{2}$ generators of each type, when $L$ is always odd. Figure 10 presents both code geometries.

Two standard noise models are examined: *independent noise* and *depolarizing noise*. Under the independent (uncorrelated) noise model, bit-flip ($X$) and phase-flip ($Z$) errors occur independently with equal error probability, allowing the decoding of $X$ and $Z$ syndromes to be treated separately. In contrast, the depolarizing noise model assigns equal probability $p/3$ to the non-identity Pauli operators, i.e., $\Pr(X) = \Pr(Z) = \Pr(Y) = \frac{p}{3}, \Pr(I) = 1 - p$, where $Y = iXZ$.

## D  TRAINING DETAILS

Our training methodology randomly samples noise within the physical error rate testing range to ensure robust generalization across different noise regimes. The model architecture employs $N = 6-8$ transformer layers with shared parameters across dual token streams and an embedding dimension of $d = 128$ and $h = 16$ attention heads. The multi-component loss function uses weighting parameters $\lambda_{\text{LP}} = 0.2$, $\lambda_{\text{LC}} = 1.0$, and $\lambda_{\text{Ent}} = 1.0$ for informed logical priors loss, logical class prediction loss, and minimum entropy loss, respectively.

We optimize using the Adam optimizer (Kingma & Ba, 2014) with mini-batches of $128 - 512$ samples over $200 - 600$ epochs, processing $5,000 - 20,000$ mini-batches per epoch for a total of

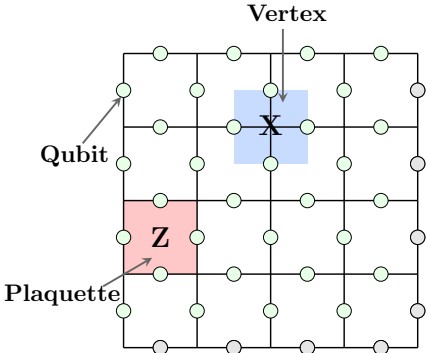 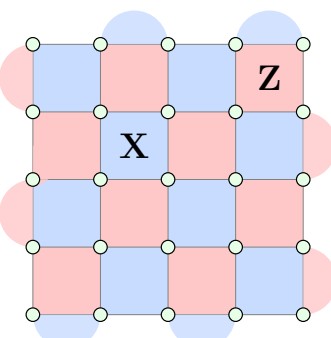

Figure 10: **Surface codes**: (left) Toric code with $L_{\text{code}} = 4$, where gray qubits represent boundary conditions with periodic boundary conditions (top row connects to bottom row, left column connects to right column). (right) Rotated surface code with $L_{\text{code}} = 5$. Data qubits adjacent to red faces correspond to Z-type stabilizer generators, while those adjacent to blue faces correspond to X-type stabilizer generators.

approximately $2.56 \times 10^6$ error samples per training run. The initial learning rate is set between $3 \times 10^{-4}$ and $1 \times 10^{-4}$, with cosine annealing decay to $1 \times 10^{-6}$ by the end of training (Loshchilov & Hutter, 2016). Detailed experimental configurations are presented in Table 2, with all experiments conducted on a 48GB NVIDIA L40S GPU.

We initialized our development from the QECCT implementation. While longer training and alternative configurations may yield further improvements, time and computational constraints limited our exploration of the hyperparameter space. We use the toric code implementation from Krastanov & Jiang (2017), while the rotated surface code is implemented from scratch.

To simulate circuit-level noise, we utilized the stim package. Quantum circuits were constructed using `stim.Circuit.generated`, with the noise parameters `after_clifford_depolarization`, `before_round_data_depolarization`, and `before_measure_flip_probability` all set to the physical error rate, $p$. The `after_reset_flip_probability` parameter was explicitly set to zero. To generate the decoding dataset, we extracted the detector error model (DEM) via `circuit.detector_error_model`.

For the Minimum Weight Perfect Matching (MWPM) baseline, we employ the PyMatching library Higgott (2022). For the Belief Propagation with Ordered Statistics Decoding (BP-OSD) decoder, we utilize the ldpc Python package Roffe (2022). The Belief Propagation stage is configured with the product-sum algorithm using a serial schedule and a maximum of 100 iterations. This is followed by an Exhaustive Ordered Statistics Decoding (OSD-E) post-processing stage with order 2, initialized with a prior channel error probability derived from the mean of the tested physical error rates.

Table 2: Experimental configuration across different code distances and noise models.

| Code Distance | Code Type | Noise Type | Learning Rate | Epochs | Batch Size | SAQ-Decoder Layers | Model Params | Physical Error Rate | Epoch Time [sec] |
|---|---|---|---|---|---|---|---|---|---|
| 4 | Toric | Independent | 2.5e-4 | 200 | 512 | 6 | 1.2M | 0.01-0.20 | 192 |
| | | Depolarizing | 2e-4 | 200 | 128 | 6 | 1.2M | 0.05-0.20 | 283 |
| 5 | Rotated Surface | Independent | 1e-4 | 200 | 128 | 6 | 1.2M | 0.01-0.20 | 153 |
| | | Depolarizing | 2e-4 | 300 | 128 | 6 | 1.2M | 0.05-0.20 | 236 |
| 5 | Color | Circuit | 2e-4 | 200 | 512 | 6 | 1.23M | 0.001-0.02 | 217 |
| 6 | Toric | Independent | 2.5e-4 | 400 | 512 | 6 | 1.2M | 0.01-0.20 | 184 |
| | | Depolarizing | 3e-4 | 400 | 512 | 6 | 1.23M | 0.05-0.20 | 478 |
| 7 | Rotated Surface | Independent | 1e-4 | 500 | 512 | 6 | 1.2M | 0.01-0.20 | 245 |
| | | Depolarizing | 3e-4 | 500 | 512 | 6 | 1.21M | 0.05-0.20 | 584 |
| 7 | Color | Circuit | 2e-4 | 200 | 512 | 6 | 1.29M | 0.001-0.02 | 438 |
| 8 | Toric | Independent | 1.5e-4 | 600 | 128 | 6 | 1.22M | 0.01-0.20 | 386 |
| | | Depolarizing | 2e-4 | 600 | 128 | 8 | 1.7M | 0.05-0.20 | 2464 |
| 9 | Rotated Surface | Independent | 1e-4 | 600 | 512 | 6 | 1.2M | 0.01-0.20 | 285 |
| | | Depolarizing | 1e-4 | 600 | 128 | 8 | 1.63M | 0.05-0.20 | 641 |
| 10 | Toric | Independent | 3e-4 | 600 | 512 | 6 | 1.26M | 0.01-0.20 | 683 |
| | | Depolarizing | 2e-4 | 600 | 512 | 8 | 1.85M | 0.05-0.20 | 3090 |
| 11 | Rotated Surface | Independent | 1e-4 | 600 | 512 | 6 | 1.21M | 0.01-0.20 | 471 |
| | | Depolarizing | 1e-4 | 600 | 512 | 8 | 1.71M | 0.05-0.20 | 1282 |

# E    COMPARISON WITH LEADING NEURAL DECODERS

This appendix provides a detailed comparison of the SAQ-Decoder's performance threshold under depolarizing noise against recent neural decoders and high-performance classical decoders. Our numerical comparison in Table 3, the results confirm that the SAQ-Decoder's achieved threshold of $18.6\%$ approaches the theoretical Maximum Likelihood (ML) bound of $18.9\%$ (Bombin et al., 2012) and represents the highest value reported for a practical, scalable decoder.

Table 3: Comparative Thresholds under Depolarizing Noise

| Method | Toric | Rotated |
|---|---|---|
| **SAQ-Decoder (Ours)** | **18.6%** | **18.3%** |
| QECCT (Choukroun & Wolf, 2024) | 17.8% | 17.2% |
| Astra (Maan & Paler, 2025) | – | 17.0% |
| SU-NetQD (Zhang et al., 2025) | 16.3% | – |
| ML+UF (Meinerz et al., 2022) | 16.2% | – |
| UIUF (Lin & Lai, 2025) | 15.5% | 15.6% |

To rigorously evaluate the performance of the proposed SAQ decoder, we benchmark it against the latest advancements in neural decoding architectures. The field has recently moved towards sophisticated deep learning models that aim to exploit the geometric structure of topological codes better than standard generic networks. In this section, we compare our method with two leading state-of-the-art approaches: the vision-transformer-based QuantumSMoE (Nguyen et al., 2026) and the U-Net-style SU-NetQD (Zhang et al., 2025). Our analysis demonstrates that SAQ consistently achieves superior logical error suppression compared to these strong baselines. By effectively modeling both local stabilizer constraints and global error patterns, the SAQ decoder establishes a new performance standard, particularly in the low-error regimes essential for practical fault-tolerant quantum computation.

Recently, QuantumSMoE has emerged as a promising decoding framework that effectively adapts vision transformer architectures to the geometric constraints of topological codes. This method explicitly incorporates code structure through specialized plus-shaped embeddings that capture local stabilizer interactions, adaptive masking to enforce lattice connectivity, and a Soft Mixture-of-Experts layer that enhances model capacity through sparse, conditional computation. While QuantumSMoE demonstrates the utility of spatial inductive biases, our proposed SAQ decoder achieves superior logical error suppression across the examined error regimes, as shown in Table 4. For instance, on the $L_{code} = 6$ toric code at a physical error rate of $p = 0.09$, SAQ reduces the logical error rate to $3.63 \times 10^{-2}$ compared to $4.92 \times 10^{-2}$ for QuantumSMoE, demonstrating robust performance improvements in the critical low-error regime and suggesting that SAQ more effectively captures the global topological correlations necessary for high-fidelity decoding.

This performance advantage extends to comparisons with other leading architectures, such as SU-NetQD. As summarized in Table 5, the proposed SAQ decoder consistently achieves the lowest logical error rate among all considered decoders across toric code with distances $L_{code} \in \{5, 7\}$ and physical error rates $p \in \{0.09, 0.13, 0.20\}$. Compared to the SU-NetQD architecture, SAQ yields systematically smaller LERs, while the classical BP-OSD-2 and MWPM decoders exhibit substantially higher error rates in the same regime. In particular, for distance $L_{code} = 7$ at $p = 0.09$, SAQ attains a logical error rate of $1.95 \times 10^{-2}$, whereas SU-NetQD, BP-OSD-2, and MWPM reach $2.76 \times 10^{-2}, 7.20 \times 10^{-2}$, and $6.90 \times 10^{-2}$, respectively. This corresponds to a relative LER reduction of about $29\%$ with respect to SU-NetQD, and roughly $73\%$ and $72\%$ reductions compared to the BP-OSD-2 and MWPM baselines, respectively, highlighting the substantial performance advantage of the proposed SAQ decoder.

# F    DETAILED COMPUTATIONAL METRICS

This appendix provides the detailed numerical comparisons of computational complexity and efficiency metrics for the SAQ-Decoder against the QECCT baseline. Metrics include the total number of floating-point operations (FLOPs), the total number of trainable parameters (Params), and the

Table 4: LER comparison of SAQ, QuantumSMoE, and baseline decoders for toric code with distance $L_{code} = 6$ at selected physical error rates $p$.

| | $p$ | SAQ | QuantumSMoE | BP-OSD-2 | MWPM |
|---|---|---|---|---|---|
| | 0.09 | **0.0363** | 0.0492 | 0.1143 | 0.1238 |
| | 0.11 | **0.0812** | 0.0985 | 0.2010 | 0.2161 |
| $L_{code} = 6$ | 0.13 | **0.1555** | 0.1690 | 0.3118 | 0.3343 |
| | 0.15 | **0.2489** | 0.2560 | 0.4216 | 0.4443 |

Table 5: LER comparison of SAQ ,SU-NetQD, and baseline decoders for toric code with distances $L_{code} = 5, 7$ at selected physical error rates $p$.

| | $p$ | SAQ | SU-NetQD | BP-OSD-2 | MWPM |
|---|---|---|---|---|---|
| | 0.09 | **0.051** | 0.057 | 0.107 | 0.108 |
| $L_{code} = 5$ | 0.13 | **0.177** | 0.191 | 0.280 | 0.279 |
| | 0.20 | **0.519** | 0.534 | 0.623 | 0.624 |
| | 0.09 | **0.019** | 0.028 | 0.072 | 0.069 |
| $L_{code} = 7$ | 0.13 | **0.139** | 0.149 | 0.262 | 0.253 |
| | 0.20 | **0.548** | 0.581 | 0.680 | 0.676 |

inference time per sample (Time). Our numerical comparison in Table 6 consistently demonstrates the superior efficiency and scalability of the SAQ-Decoder across all tested code distances ($L$) and noise models (independent 'ind' and depolarizing 'dep').

## G  THE USE OF LARGE LANGUAGE MODELS (LLMS)

LLMs were employed to assist in several aspects of this research and manuscript preparation. For the literature review, LLMs aided in the identification and sourcing of relevant related works to ensure comprehensive coverage of the field. During the research process, LLMs were consulted for research ideation, though these explorations did not yield beneficial outcomes that influenced the final work. In manuscript preparation, LLMs assisted with improving grammar, enhancing textual transitions between sections, and refining the exposition of technical concepts for better clarity and readability. For software development, LLMs provided assistance in code writing, GPU acceleration optimizations, and debugging code issues. Despite these auxiliary uses, all core research contributions, experimental design, theoretical insights, and scientific conclusions presented in this work are entirely the product of the authors' original research and analysis.

Table 6: Comparison of SAQ and QECCT Decoder Computational Metrics

| Code Type | $L_{code}$ | Noise | Decoder | FLOPs [M ↓] | Params [M ↓] | Time [sec ↓] |
|---|---|---|---|---|---|---|
| **Toric** | 4 | ind | SAQ | **27.48** | **1.20** | **2.13e-05** |
| | | | QECCT | 57.01 | 1.23 | 2.61e-05 |
| | | dep | SAQ | **61.51** | **1.20** | **2.59e-05** |
| | | | QECCT | 114.09 | 1.33 | 8.13e-05 |
| | 6 | ind | SAQ | **55.16** | **1.20** | **2.27e-05** |
| | | | QECCT | 128.37 | 1.37 | 1.01e-04 |
| | | dep | SAQ | **116.88** | **1.23** | **6.39e-05** |
| | | | QECCT | 257.08 | 1.90 | 3.41e-04 |
| | 8 | ind | SAQ | **93.92** | **1.22** | **5.01e-05** |
| | | | QECCT | 228.45 | 1.75 | 2.73e-04 |
| | | dep | SAQ | **259.15** | **1.70** | **2.15e-04** |
| | | | QECCT | 458.01 | 3.42 | 9.72e-04 |
| | 10 | ind | SAQ | **143.76** | **1.26** | **9.72e-05** |
| | | | QECCT | 357.44 | 2.55 | 6.28e-04 |
| | | dep | SAQ | **392.09** | **1.85** | **4.50e-04** |
| | | | QECCT | 717.61 | 6.64 | 2.33e-03 |
| **Rot. Surf.** | 5 | ind | SAQ | **19.97** | **1.20** | **6.63e-06** |
| | | | QECCT | 43.94 | 1.21 | 1.78e-05 |
| | | dep | SAQ | **38.55** | **1.20** | **1.50e-05** |
| | | | QECCT | 87.92 | 1.28 | 5.49e-05 |
| | 7 | ind | SAQ | **36.57** | **1.20** | **1.46e-05** |
| | | | QECCT | 86.73 | 1.27 | 5.37e-05 |
| | | dep | SAQ | **71.77** | **1.21** | **3.23e-05** |
| | | | QECCT | 173.62 | 1.52 | 1.75e-04 |
| | 9 | ind | SAQ | **58.72** | **1.20** | **2.35e-05** |
| | | | QECCT | 143.84 | 1.41 | 1.23e-04 |
| | | dep | SAQ | **154.71** | **1.63** | **9.15e-05** |
| | | | QECCT | 288.13 | 2.08 | 4.16e-04 |
| | 11 | ind | SAQ | **86.40** | **1.22** | **4.49e-05** |
| | | | QECCT | 215.33 | 1.69 | 2.51e-04 |
| | | dep | SAQ | **228.54** | **1.68** | **1.73e-04** |
| | | | QECCT | 431.66 | 3.18 | 8.87e-04 |

