# OpenReview forum: "SAQ: Stabilizer-Aware Quantum Error Correction Decoder"
_ICLR.cc/2026/Conference — ICLR 2026 Poster_

### Official Review · Reviewer_xQqG · 2025-10-29

**Soundness:** 4
**Presentation:** 4
**Contribution:** 4
**Rating:** 4
**Confidence:** 1

**Summary:**

I would like to note that I am not familiar with the quantum computing or quantum machine learning domain, and therefore I do not have the necessary background to properly evaluate this submission. While I can assess general aspects of clarity and structure, I am not qualified to judge the technical novelty, correctness, or significance of the paper within the context of quantum research.

I respectfully suggest that this paper be reassigned to a reviewer with expertise in quantum algorithms or quantum information, to ensure a fair and accurate evaluation.

**Strengths:**

See Summary

**Weaknesses:**

See Summary

**Questions:**

See Summary

---

### Official Review · Reviewer_wwv5 · 2025-10-29

**Soundness:** 3
**Presentation:** 4
**Contribution:** 3
**Rating:** 4
**Confidence:** 3

**Summary:**

The paper introduces SAQ-Decoder. It is a neural network approach for QEC that addresses the fundamental accuracy efficiency trade off in QEC decoding. The method combines a dual-stream transformer architecture with a novel differentiable logical loss function and a post-processing algorithm, constraint projected null space descent. The approach achieves near optimal error thresholds while maintaining linear computational complexity.

**Strengths:**

1.  Strong empirical performance. The decoder achieves very good results on toric codes, also approaching maximum bounds while outperforming both classical methods and recent neural approaches.
2. Novel architectural design. The dual-stream transformer with asymmetric attention patterns is well-motivated for QEC.
3. Differentiable logical loss. Paper provides a rigorous mathematical derivation of a differentiable approximation to the discrete GF(2) constraints, enabling end to end training that directly optimizes logical error rates rather than bit error rates.

**Weaknesses:**

1. The experiment limited to code distances up to 10. For practical fault-tolerant quantum computing should be larger.
2. The paper lacks comparison with recent strong baselines.
3. While the method clams applicability to any stabilizer code family, but experiments are limited to surface codes.

**Questions:**

1. The ablation studies don’t clearly show CPND’s contribution versus simpler projection. Can you provide a direct comparison of logical error rates with/without CPND?
2. Does decoder require retraining for different physical error models?
3. What are the convergence guarantees for CPND?

---

> ### Author Response · Authors · 2025-11-16
> **Response to Reviewer wwv5 (part 1)**
>
> We sincerely thank the reviewer for their very positive and insightful review. We are especially encouraged that they found our paper's presentation to be "excellent," the empirical performance "strong," and our architectural design and logical loss to be "novel," "well-motivated," and "rigorous."
>
> We were also very grateful for the clear and actionable weaknesses identified. The reviewer correctly pointed out that our initial experiments were limited in code distance, baseline comparisons, and code family.
>
> We are pleased to report that we have performed significant new experiments that directly address all three of these weaknesses. We have new results for larger code distances (L=11), new experiments on entirely different code families (Color and Repetition codes) under circuit-level noise.
>
> We believe these new additions substantially strengthen the paper, and we will now address each point in detail.
>
> ## **Weakness 1**
>
> We thank the reviewer for this critical point. We agree that demonstrating scalability is essential.
>
> To provide concrete evidence of scalability, we have successfully trained and evaluated our decoder for the rotated surface code with $L=11$ for both independent and depolarizing noise models.
>
> These new results, have been placed (with plots) in the supplementary file: **supp_ICLR_2026_rebuttal.pdf**  (figure 14).
>
> As demonstrated in the figure 14, our SAQ-Decoder scales effectively, maintaining its significant performance gap over both MWPM and QECCT and thus confirming our method's robustness for larger, more practical code distances.
>
> We are already running additional experiments for $L > 11$. As these require significant training time, we will incorporate these new scalability results into the final version of the paper.
>
> ## **Weakness 2**
>
> We thank the reviewer for this important point, as it allows us to further contextualize our contribution. Our SAQ-Decoder achieves an error threshold of 18.6%/18.3% for toric/rotated-surface codes under depolarizing noise. To our knowledge, this is the highest threshold reported for a practical, scalable decoder, approaching the theoretical Maximum Likelihood (ML) bound of 18.9%.
>
> To address the reviewer's concern, we have compiled a comparison of SAQ-Decoder thershold under depolarization noise in surface code against the recent strong baselines the reviewer is likely referring to, including SOTA neural decoders (GNNs, U-Nets) and high-performance classical decoders. This broader comparison confirms that SAQ-Decoder's performance is state-of-the-art.
>
> | Decoder / Paradigm               | Reported Threshold (Depolarizing Noise) |
> |----------------------------------|-----------------------------------------|
> | SAQ-Decoder (Our Work)             | **18.6%**                               |
> | QECCT (Transformer Baseline) [1]   | 17.8%                                   |
> | Astra (2024)                 [2]   | ~17%                                    |
> | SU-NetQD (2025)              [3]   | 16.3%                                   |
> | ML+UF (2022)                 [4]   | 16.2%                                   |
> | MWPM (Classical Baseline)    [5]   | 16%                                     |
> | BP-OSD (Classical Baseline)  [6]   | ~16%                                    |
> | UIUF (2024)                  [7]   | 15.6%                                   |
>
> We will add this expanded comparative analysis to the final version of the paper. We would also be very grateful if the reviewer could suggest any other specific baselines they believe are essential for comparison.
>
>
> ### References:
> - [1] Choukroun & Wolf, "Deep quantum error correction", AAAI (2024).
> - [2] Arshpreet Singh Maan and Alexandru Paler, "Machine Learning Message-Passing for the Scalable Decoding of QLDPC Codes", (2024).
> - [3] Zhang, Wei-Wei and Xia, Zhuo and Zhao, Wei and Pan, Wei and Shi, Haobin, "Self-attention U-Net decoder for toric codes", Physical Review Applied (2025).
> - [4] Meinerz, Kai and Park, Chae-Yeun and Trebst, Simon, "Scalable Neural Decoder for Topological Surface Codes", Physical Review Letters (2022).
> - [5] D. S. Wang and A. G. Fowler and A. M. Stephens and L. C. L. Hollenberg, "Threshold error rates for the toric and surface codes", (2009).
> - [6] ldpc package (BP-OSD Decoder), our findings.
> - [7] Lin & Lai, "Union-Intersection Union-Find for Decoding Depolarizing Errors in Topological Codes", IEEE Journal on Selected Areas in Information Theory (2025).

---

> > ### Author Response · Authors · 2025-11-16
> > **Response to Reviewer wwv5 (part 2)**
> >
> > ## **Weakness 3**
> >
> > We thank the reviewer for highlighting this. Our claim of generality was based on the framework's design, which is fundamentally agnostic to the code family. To experimentally validate this generality, we have successfully run new experiments on two entirely different code families under realistic circuit-level noise:
> > 1. **Color Code** (L=3,5 with 2 rounds of circuit noise)
> > 2. **Repetition Code** (L=3,5 with 3 rounds of circuit noise)
> >
> > We add these comparisons to our experiments in  the supplementary file: **supp_ICLR_2026_rebuttal.pdf** (figures 15-16).
> >
> > Our decoder demonstrated strong performance in both cases, proving it is not specialized to surface codes and can handle complex, circuit-level noise models. We have will add these significant new results, including plots of LER vs. physical error rate. This new evidence confirms that SAQ-Decoder is a truly generalizable framework applicable to diverse stabilizer code. In particular, for the Color Code (L=5) at $p=2.00e-02$, our SAQ-Decoder is 17.0% better than QECCT and 64.2% better than MWPM.
> >
> > ## **Question 1**
> >
> > We thank the reviewer for this observation. The LER of simpler projection (our CPND's first stage) and the full CPND is identical. The simpler projection already enforces the syndrome and logical constraints (i.e., enforcing the CPND's output to hold the logical class as was predicted by the logical-stream).
> >
> > To clarify, the 'SAQ-decoder without CPND' is the raw error prediction $\hat{e}$ generated by our Syndrome Stream, which is directly analogous to the final $\hat{\epsilon}$ output of QECCT. We have performed this analysis and found that our raw $\hat{e}$ prediction, even without post-processing, already achieves a significantly lower LER than the final QECCT decoder. We will add this new comparison to our experiments.
> >
> > We add a new comparison to our experiments in the supplementary file: **supp_ICLR_2026_rebuttal.pdf** (figures 10-13).
> >
> > Crucially, the performance of this "SAQ (No CPND)" model is not far from our full, CPND-enabled SAQ decoder, while maintaining a high marginal performance gap over all other baselines. In particular, **Figure 10c** ($L=10$ toric code under depolarizing noise) shows that at $p = 0.2$, taking SAQ (LER $= 6.06 \times 10^{-1}$) as a reference, MWPM and QECCT incur LER increases of approximately $26\%$ and $19\%$, respectively, whereas `SAQ (No CPND)` increases the LER by only $2.8\%$. This indicates that `SAQ (No CPND)` suffers only a very small marginal degradation relative to SAQ, while maintaining the high marginal gap from the classical and neural baselines.
> >
> >
> > ## **Question 2**
> >
> > Yes, the decoder is trained for a specific noise model. This specialization allows it to learn the precise statistics of that noise, which is key to its high performance. Our new experiments confirm this approach is highly effective, showing the decoder also achieves strong results on different code families under realistic, structured circuit-level noise.
> >
> > ## **Question 3**
> > We thank the reviewer for this excellent question. The guarantee of termination at a local minimum comes from three key properties working together:
> > 1. **Finite Search Space:** The algorithm is searching within a finite set of possible solutions (the affine space $e' \oplus \ker(\hat{H})$). The number of valid recovery operators is finite.
> > 2. **Monotonic Cost Reduction:** The algorithm is greedy. It only makes a move (flipping the error estimate by a nullspace generator $v_j$) if that move strictly decreases the total weighted cost (i.e., $\Delta_j < 0$, as shown in Algorithm 1). The cost can only go down, it never increases.
> > 3. **Deterministic Process:** The algorithm checks each generator in a fixed, single pass.
> >
> > Because the search space is finite and the cost is always decreasing with every move with finite number of steps (the null space dimention, which propotional to the squared code distance), the algorithm cannot get stuck in a loop or run forever, it must eventually stop. It stops at a local minimum because the process terminates when it finds a solution $e'$ where no single generator $v_j$ in its list can be applied to decrease the cost any further. It's "local" because it has found the best solution in its immediate "neighborhood" (as defined by single generator flips), not necessarily the global best solution

---

> ### Comment · Reviewer_wwv5 · 2025-11-26
>
> I found the authors have partially addressed my review comments and questions, however, the over-claimed points are still valid in the current presentation of this work. So I decided to to maintain my current score.

---

> > ### Author Response · Authors · 2025-11-26
> > **Response to Reviewer wwv5**
> >
> > We thank the reviewer for their final comment and valuable engagement. We recognize the importance of ensuring that our claims are precise and perfectly aligned with the presented technical details and results. We believe the reviewer's concern regarding 'over-claimed points' pertains to the language we used when describing the novelty and general applicability of the framework.
> >
> > To specifically address the remaining issue, could the reviewer kindly indicate which specific statements or sections in the current version of the manuscript (including the appendices detailing the rebuttal content) they still consider "over-claimed"? Identifying these specific points will allow us to immediately implement precise linguistic revisions to ensure accuracy and meet the reviewer's standard of presentation.
> >
> > Given the experiments added to the rebuttal that addressed the weaknesses (scalability to $L_{code}=11$, generality across codes like Color and Repetition codes with circuit noise, and inclusion of the strong BP-OSD2 baseline ), we are ready to perform any additional clarifying experiments or provide supplementary analysis if the reviewer requires more evidence to fully demonstrate the performance, scalability, or generality of the SAQ-Decoder. Could the reviewer suggest any specific remaining clarifications or additional experiments needed to fully satisfy the review criteria?
> >
> > Thank you again for your time and guidance in improving this work.

---

> > > ### Author Response · Authors · 2025-12-01
> > > **Response to Reviewer wwv5**
> > >
> > > We thank the reviewer for their final comment. We take the concern regarding "over-claimed points" very seriously. We recognize that while our rebuttal provided extensive new empirical evidence (including Color Codes, Repetition Codes, $L_{code}=11$, and BP-OSD comparisons), the manuscript text required precise refinement to ensure our claims strictly matched this new experimental scope. To address the reviewer's concern and ensure maximum scientific rigor, we have revised the manuscript to explicitly scope our claims of generality and scalability.
> > >
> > > 1. **Refined Claim of Generality:** We acknowledge that claiming applicability to *"any stabilizer code"* was too broad given the primary focus on surface codes. We have revised the Introduction to scope our contribution to adaptability *"across lattice geometries"*, a claim now directly supported by our new experiments on Color and Repetition codes (Figure 6).
> > > 2. **Refined Claim of Scalability:** To address concerns regarding complexity and code distance, we have qualified our scalability claims to be mathematically precise regarding the input size, while providing empirical proof of scaling up to Rotated Surface Code $L_{code}=11$ (Figure 4).
> > > 3. **Refined Claim of Performance:** To avoid any ambiguity regarding *"optimality"*, we have refined our abstract to state that the decoder "closely approaches theoretical Maximum Likelihood bounds" (e.g., 18.6% vs. 18.9% for Toric codes ). This replaces general claims of optimality with quantitative facts.
> > >
> > > We believe these revisions, combined with the new BP-OSD baselines and the validation on Color Codes and Repetition Codes, resolve the concern of "over-claiming." The manuscript now presents a precise, empirically validated framework for topological quantum error correction.
> > >
> > > We once again thank the reviewer for their time and valuable guidance in improving this work.

---

### Official Review · Reviewer_ehw9 · 2025-10-31

**Soundness:** 3
**Presentation:** 2
**Contribution:** 2
**Rating:** 4
**Confidence:** 4

**Summary:**

The authors propose a transformer-based quantum decoder, called the SAQ-decoder, with several innovative features: dual-stream transformers, a differentiable logical loss, and constraint-aware post-processing. The proposed decoder achieves near-optimal accuracy for topological codes.

**Strengths:**

The dual-stream transformer architecture is intriguing, and the experimental results appear very promising.

**Weaknesses:**

1.	It’s unclear what the final output of the proposed decoder actually is. Is it \hat{l}, \hat{e}, or the output of the CPND? Based on the appendix, it seems the final output used for evaluating LER is from the CPND, but this should be clarified more clearly in the main text. Also, e^{pred} above Eq. (13) appears to represent the final output, but this is not explicitly stated.
2.	To my knowledge, other transformer-based decoders for QEC already exist, such as alphaQubit. These should be cited, and the differences explained. Additionally, the dual-stream transformer architecture seems similar to CrossMPT for classical codes, which also uses cross-attention. That work is worth referencing as well.
3.	The MLP ϕb_\phi seems technically identical to the FFN in Kai (2022, PRL), as it takes the syndrome vector and outputs logical information. This similarity should be acknowledged.
4.	The asymmetric structure of the dual streams is not intuitive. Why does the syndrome stream (which contains local information) influence the logical stream (which contains global information)? The opposite direction seems more meaningful. Without empirical results, can the authors illustrate the reasoning behind this design? Moreover, since the syndrome stream is not affected by the logical stream, does this mean the error vector is derived directly from the input syndrome? If so, how does this differ from QECCT?
5.	Is the gain over QECCT mainly due to the novel transformer architecture or the post-processing via CPND? To make a fair comparison, the SAQ-decoder should also be compared to QECCT without CPND.
6.	The non-differentiability of the loss function L(e_{\text{true}} + e_{pred}) has already been addressed by Liu (2019, PRL). Could you explain the difference?
7.	The authors use MWPM as the standard classical decoder. However, BP+OSD is also a well-known and widely used decoder for quantum codes and should be included in the comparison.
8.	The computational complexity is only discussed using Big-O notation. It would be more informative to also provide numerical comparisons, such as FLOPs or inference time.
9.	Regarding parameter efficiency, the SAQ-decoder still relies on syndrome information, which scales quadratically with the lattice size LLL. Therefore, I think it cannot avoid quadratic scaling.

**Questions:**

In Fig. 6, are the authors testing the effect of both masking and the global token? Does the "no mask" label mean only the global token is used, and "mask only" means only masking is applied? Is SAQ-decoder the version with both features? The labeling is somewhat confusing. Also, it would be helpful to include results where neither masking nor the global token is used.

---

> ### Author Response · Authors · 2025-11-16
> **Response to Reviewer ehw9 (part 1)**
>
> We sincerely thank the reviewer for their exceptionally thorough and constructive feedback. We are encouraged that they found our dual-stream architecture "intriguing" and our experimental results "very promising."
>
> We are particularly grateful for their detailed critique of our paper's presentation and the list of missing citations. This feedback provided a clear roadmap for improvement, which we have followed closely in our revision.
>
> We will now address every weakness and question in detail:
>
> ## **Weakness 1**
> We thank the reviewer for pointing out this lack of clarity. We will revise the main text to explicitly detail the information flow of our decoder, as this is a key part of our hybrid design. The reviewer is correct on all points:
> 1. The final output used for evaluating LER is indeed the recovery operator $e(s)$, which is the output of the CPND stage.
> 2. The transformer outputs, $\hat{l}$ and $\hat{e}$, are inputs to this CPND stage. We will update the main text in Section 4 to explicitly describe this sequential flow: (1) The SLTD transformer outputs the logical logits $\hat{l}$ and the error logits $\hat{e}$. (2) These are passed to the CPND algorithm. (3) The binarized $\hat{l}$ is used to set the logical constraint $l$ (part of the vector $b$), and the logits $\hat{e}$ are used to define the reliability weights $w_q$.
> 3. The term $e^{\text{pred}}$ (noted by the reviewer above Eq. 13) is the hard-decision on the $\hat{e}$ logits. This serves as the initial "prior" prediction which is fed into the CPND algorithm (as $e^{\text{pred}}$ in Algorithm 1). CPND then algorithmically refines this $e^{\text{pred}}$ to produce the final $e(s)$ that is guaranteed to be syndrome-consistent.
>
> We will also clarify that the $e^{\text{pred}}$ is distinct from the final inference output $e(s)$.
>
>
>
> ## **Weakness 2**
> We thank the reviewer for these relevant citations, which we will add to our revised related work section in the paper.
>
> **AlphaQubit (Bausch et al., 2024)** is indeed a significant transformer-based decoder, but it differs from our SAQ-Decoder in several crucial aspects:
> 1. Architecture: AlphaQubit is a recurrent model (RNN + Transformer) designed to process syndrome data round-by-round and maintain an internal "decoder state". Our SAQ-Decoder is a feed-forward, dual-stream architecture that processes the entire syndrome in a single pass.
> 2. Mechanism: AlphaQubit is a single, end-to-end neural network that directly predicts the final logical error. SAQ-Decoder is a hybrid neural-algorithmic framework. Its two streams output a logical class prediction ($\hat{l}$) and physical error predictions ($\hat{e}$), which are then fed into the deterministic CPND algorithm to construct a guaranteed syndrome-consistent solution.
> 3. Input Data: A core feature of AlphaQubit is its design to utilize rich, analogue inputs, such as "soft readouts" (I/Q values) and "leakage information". Our SAQ-Decoder, in contrast, is designed to operate on the binary syndrome vector.
> 4. Training: AlphaQubit relies on a two-stage approach of pretraining and finetuning, to adapt to real experimental data. Our work employs a standard, single-stage supervised training methodology.
>
> **CrossMPT (Park et al., 2024)**: Regarding CrossMPT, we agree with the reviewer that it is a highly relevant work in the classical coding domain. Our related work section was intentionally focused on the quantum error correction (QEC) literature, primarily due to severe space constraints.  We will add this citation in our revision.
>
> ## **Weakness 3**
> We thank the reviewer for highlighting the relevant work by Kai (2022, PRL), the similarity will be acknowledged.
> We will clarify the key functional distinction in our revision. The FFN in Meinerz et al. acts as a local pre-decoder, mapping a small $l \times l$ syndrome patch to the error probability of the single central qubit.
> In contrast, our MLP $b_\phi$ performs a global estimation, mapping the entire syndrome vector $s$ to an initial estimate of the global logical class $\tilde{l}$.
> This estimate is then used as an informed prior to construct the logical token stream for our transformer, rather than as a direct correction.

---

> > ### Author Response · Authors · 2025-11-16
> > **Response to Reviewer ehw9 (part 2)**
> >
> > ## **Weakness 4**
> >
> > We thank the reviewer for this insightful question, which allows us to clarify the core principles of our design and its distinction from QECCT.
> > 1. Intuition of the Asymmetric Structure: The intuition that global information (logical) should influence local information (syndrome) is an interesting one. However, our design is based on the reverse information flow: local evidence is integrated to form a global conclusion.
> >    * In our framework, the Syndrome Stream ($T_S$) represents the set of local, physical "evidence" (the syndrome violations).
> >    *  The Logical Stream ($T_L$) represents the "global hypothesis" (the logical error class).
> >
> >     Our asymmetric cross-attention models the process of the global hypothesis (logical tokens) "looking at" all the local evidence (syndrome tokens) to refine itself and reach a final, globally-informed conclusion. The Syndrome Stream's processed information thus serves to inform and improve the primary task of logical prediction.This design choice is also supported by our empirical results. In **Figure 6a** (Architecture study), we tested a "Bidirectional Cross-Attn" variant where both streams attend to each other. This symmetric model performed worse than our asymmetric SAQ-Decoder architecture, as our asymmetric structure ("SAQ-Decoder") reduces the LER by 11.4% compared to the symmetric model ("Bidirectional Cross-Attn")
> > 3. Difference from QECCT: It is correct that our final physical error prediction $\hat{e}$ is derived from the Syndrome Stream. However, our architecture and decoding pipeline are fundamentally different from QECCT, particularly in what this $\hat{e}$ is used for.
> >     * QECCT: Uses a single-stream architecture. Its one and only output is the final physical error vector $\hat{e}$. This output is the final answer and is not guaranteed to be syndrome-consistent.
> >     * SAQ-Decoder: Ours is a hybrid, multi-stage framework with decoupled tasks:
> >       1. Logical Coset Prediction (The LER Driver): Our Logical Stream is designed to produce a direct prediction of the logical coset, $\hat{l}$. This prediction is the primary driver of the LER performance. Our ablation study in **Figure 6b** confirms that the logical classification loss ($\mathcal{L}_{LC}$) is the critical component for this task, contributing to a 7.2% reduction in the LER.
> >       2. Error Prediction (A Prior for CPND): Our Syndrome Stream produces a separate error prediction, $\hat{e}$. This output, which is analogous to the entire output of QECCT, does not directly determine the LER in our framework.
> >       3. Algorithmic Post-processing: Both $\hat{l}$ and $\hat{e}$ are fed into the CPND stage. CPND uses the logical prediction $\hat{l}$ to define the target logical coset and the error prediction $\hat{e}$ as a "good prior" (as reliability weights $w_q$). It then algorithmically constructs a final recovery operator $e(s)$ that guarantees two things: (1) It is syndrome-consistent with the input $s$. (2) It preserves the logical coset prediction $\hat{l}$ provided by the Logical Stream.
> >
> > To conclude, QECCT's network directly predicts the final (and potentially inconsistent) error vector. Our network predicts the logical coset (which determines LER) and provides a weighted prior ($\hat{e}$), which our CPND algorithm then uses to construct a guaranteed syndrome-consistent solution within that predicted coset.
> >
> > ## **Weakness 5**
> >
> > We thank the reviewer for this excellent question, which prompted us to perform a more direct "apples-to-apples" comparison.
> >
> > To create a direct architectural comparison, our "SAQ-decoder without CPND" experiment uses the raw error prediction $\hat{e}$ from our Syndrome Stream. As you noted, this provides a fair, "apples-to-apples" comparison, as this $\hat{e}$ is the direct equivalent of the final output $\hat{\epsilon}$ from the end-to-end QECCT model.
> >
> > We have performed this analysis and found that the LER of our raw error prediction $\hat{e}$ by itself already significantly outperforms the final LER of the QECCT decoder.
> >
> > We add a new comparison to our experiments in the supplementary file: **supp_ICLR_2026_rebuttal.pdf**  (figures 10-13).
> >
> > Crucially, the performance of this "SAQ (No CPND)" model is not far from our full, CPND-enabled SAQ decoder, while maintaining a high marginal performance gap over all other baselines, including QECCT, MWPM, and BP-OSD. In particular, **Figure 10c** ($L=10$ toric code under depolarizing noise) shows that at $p = 0.2$, taking SAQ (LER $= 6.06 \times 10^{-1}$) as a reference, MWPM, BP-OSD, and QECCT incur LER increases of approximately $26\%$, $23\%$, and $19\%$, respectively, whereas `SAQ (No CPND)` increases the LER by only $2.8\%$. This indicates that `SAQ (No CPND)` suffers only a very small marginal degradation relative to SAQ, while maintaining the high marginal gap from the classical and neural baselines.

---

> > > ### Author Response · Authors · 2025-11-16
> > > **Response to Reviewer ehw9 (part 3)**
> > >
> > > ## **Weakness 6**
> > >
> > > We thank the reviewer for this important question and for pointing us to the key work by Liu (2019, PRL), which we will add to our discussion in Section 4.4.
> > > Both papers indeed propose differentiable surrogates for $GF(2)$ constraints, but our approach differs in two fundamental ways:
> > > 1. Separation of Objectives:
> > >     * The loss function in Liu (2019, PRL)  uses the matrix $H^{\perp}$ to simultaneously penalize both logical errors and syndrome inconsistencies. It aims to find a correction $e^{\text{pred}}$ such that $e^{\text{true}} \oplus e^{\text{pred}}$ is a stabilizer.Our framework decouples these two objectives.
> > >     * Our $\mathcal{L}_{\text{Entropy}}$ only penalizes logical errors, directly optimizing the condition $L(e^{\text{true}} \oplus e^{\text{pred}}) = 0$. We train the transformer to focus exclusively on finding the correct logical coset. The syndrome consistency is then handled separately and deterministically by our CPND post-processing stage (while preserving the exact logical coset prediction). This separation allows the neural network to solve the complex logical estimation task, while the algorithmic CPND stage guarantees a valid, syndrome-consistent solution.
> > >
> > > 2. Mathematical Formulation (Analogue vs. Probabilistic):
> > >     * The loss in Liu (2019, PRL) is an analogue surrogate. It uses the function $f(x) = |\sin(\pi x/2)|$, which is chosen because its properties (i.e., its roots at even integers) align with the discrete goal of a successful $GF(2)$ parity check.
> > >     * Our $\mathcal{L}_{\text{Entropy}}$ is a probabilistically-derived objective. As detailed in our Appendix A, we do not choose an analogue function. Instead, we derive a differentiable objective from the closed-form probability of a logical error event, $Pr(L_i \cdot r = 1)$, which is calculated based on our per-qubit flip probabilities. Our loss is the negative log-likelihood of the success event ($L_i \cdot r = 0$). Thus, we are optimizing the probability of the outcome, rather than approximating the mechanism of the check with an analogue function.
> > >
> > > We will add this citation in our revision.
> > >
> > > ## **Weakness 7**
> > >
> > > We thank the reviewer for this excellent suggestion. We have now conducted a full set of new experiments, running our SAQ-Decoder against BP-OSD for all code distances and noise models presented in the paper.
> > >
> > > We add a new comparison to our experiments in the supplementary file: **supp_ICLR_2026_rebuttal.pdf**  (figures 10-13).
> > >
> > > The results show that our SAQ transformer architecture consistently and significantly outperforms BP-OSD across the entire range of tested physical error rates and cope types.

---

> ### Author Response · Authors · 2025-11-16
> **Response to Reviewer ehw9 (part 4)**
>
> ## **Weakness 8**
>
>  We thank the reviewer for this excellent suggestion.
>  We agree completely that while Big-O notation is useful for analyzing asymptotic scaling, concrete numerical comparisons are far more informative for assessing practical performance.
>
> To address this, we have conducted a detailed analysis of our decoder's computational cost and compared it against the baselines. The results are summarized in the table below, which we will add to the revised manuscript.
>
> As the data shows, our SAQ-Decoder is not only scalable in theory but also significantly more efficient in practice. It demonstrates substantially faster inference times and a much lower computational footprint (FLOPs). This is particularly evident in parameter efficiency: at $L=10$ under depolarizing noise, the QECCT model grows to 6.64M parameters, while our SAQ-Decoder is 3.5x smaller at 1.85M. We will add this table and a corresponding discussion to Section 5.2 to provide a much clearer picture of the decoder's practical efficiency.
>
> We will add this table and a corresponding discussion to Section 5.2 to provide a much clearer picture of the decoder's practical efficiency.
>
> **Table: Comparison of SAQ and QECCT Decoder Computational Metrics**
>
> | Code Type | $L$ | Noise | Decoder | FLOPs [M $\downarrow$] | Params [M $\downarrow$] | Time [sec $\downarrow$] |
> | :--- | :---: | :---: | :--- | ---: | ---: | ---: |
> | **Toric** | 4 | ind | SAQ | **27.48** | **1.20** | **2.13e-05** |
> | | | | QECCT | 57.01 | 1.23 | 2.61e-05 |
> | | | dep | SAQ | **61.51** | **1.20** | **2.59e-05** |
> | | | | QECCT | 114.09 | 1.33 | 8.13e-05 |
> | | 6 | ind | SAQ | **55.16** | **1.20** | **2.27e-05** |
> | | | | QECCT | 128.37 | 1.37 | 1.01e-04 |
> | | | dep | SAQ | **116.88** | **1.23** | **6.39e-05** |
> | | | | QECCT | 257.08 | 1.90 | 3.41e-04 |
> | | 8 | ind | SAQ | **93.92** | **1.22** | **5.01e-05** |
> | | | | QECCT | 228.45 | 1.75 | 2.73e-04 |
> | | | dep | SAQ | **259.15** | **1.70** | **2.15e-04** |
> | | | | QECCT | 458.01 | 3.42 | 9.72e-04 |
> | | 10 | ind | SAQ | **143.76** | **1.26** | **9.72e-05** |
> | | | | QECCT | 357.44 | 2.55 | 6.28e-04 |
> | | | dep | SAQ | **392.09** | **1.85** | **4.50e-04** |
> | | | | QECCT | 717.61 | 6.64 | 2.33e-03 |
> | **Rot. Surf.** | 5 | ind | SAQ | **19.97** | **1.20** | **6.63e-06** |
> | | | | QECCT | 43.94 | 1.21 | 1.78e-05 |
> | | | dep | SAQ | **38.55** | **1.20** | **1.50e-05** |
> | | | | QECCT | 87.92 | 1.28 | 5.49e-05 |
> | | 7 | ind | SAQ | **36.57** | **1.20** | **1.46e-05** |
> | | | | QECCT | 86.73 | 1.27 | 5.37e-05 |
> | | | dep | SAQ | **71.77** | **1.21** | **3.23e-05** |
> | | | | QECCT | 173.62 | 1.52 | 1.75e-04 |
> | | 9 | ind | SAQ | **58.72** | **1.20** | **2.35e-05** |
> | | | | QECCT | 143.84 | 1.41 | 1.23e-04 |
> | | | dep | SAQ | **154.71** | **1.63** | **9.15e-05** |
> | | | | QECCT | 288.13 | 2.08 | 4.16e-04 |
> | | 11 | ind | SAQ | **86.40** | **1.22** | **4.49e-05** |
> | | | | QECCT | 215.33 | 1.69 | 2.51e-04 |
> | | | dep | SAQ | **228.54** | **1.68** | **1.73e-04** |
> | | | | QECCT | 431.66 | 3.18 | 8.87e-04 |
> $L$ denotes code distance.
>
> Inference time averaged over 100,000 samples on an NVIDIA L40S GPU.
>
> We would like to add a brief clarifying note on the parameter counts. The reviewer may notice minor differences between the Params [M] in this table and the Model Params reported in our original Appendix D (Table 1). This is due to the different reporting methods; the new table above was generated using a high-precision analysis package that counts only the active, trainable parameters at inference time. The numbers in our original Appendix D were a standard programmatic printout that may have included some unused or non-trainable components.
>
> ## **Weakness 9**
>
> We thank the reviewer for this crucial observation, our phrasing was ambiguous.
> Our claim of "linear scalability" refers to the computational complexity of our decoder with respect to its input, the syndrome size.
> We will revise the paper to make this distinction clear.

---

> > ### Author Response · Authors · 2025-11-16
> > **Response to Reviewer ehw9 (part 5)**
> >
> > ## **Question 1**
> >
> > We thank the reviewer for this question and agree that the legend for Figure 6c is confusing. We will revise the legend and caption in the final manuscript to make this much clearer. The reviewer's interpretation is correct, and we are indeed testing the separate and combined effects of masking and the global token. The three curves in the plot represent the following ablations, all on the syndrome stream:
> > 1. "No Mask": This is the baseline the reviewer asked for, representing an architecture with neither masking nor the global token. In this variant, only the syndrome tokens are used, and they interact via dense, unmasked self-attention.
> > 2. "Mask Only": This variant uses only the syndrome attention mask ($H\cdot H^T + I$) to enforce topological locality, but does not include the global token.
> > 3. "SAQ-Decoder": This is our full, final model, which includes both the locality mask ($H\cdot H^T + I$) and the global aggregation token.
> >
> > Our results show that adding the mask ("Mask Only") provides a substantial benefit over the unmasked baseline ("No Mask"), and that adding the global token on top of that ("SAQ-Decoder") provides the final, best performance with a total reduction of 9.22%. We will update the legend to be more explicit, for example: "Baseline (No Mask, No Global)", "Mask Only (No Global)", and "SAQ-Decoder (Mask + Global)".

---

### Official Review · Reviewer_H6iu · 2025-11-01

**Soundness:** 3
**Presentation:** 3
**Contribution:** 3
**Rating:** 6
**Confidence:** 1

**Summary:**

SAQ-Decoder proposes a stabilizer-aware decoder for Quantum error codes, that combines existing transformer-based decoding approaches with a constraint-inducing post-processing. The key idea is to use a dual-stream representation - one for syndromes and one for logical information. These streams are then processed by a transformer, which is followed by a lightweight constraint-preserving post-processing stage that enforces exact syndrome consistency.

The architecture attains near-ML thresholds on toric codes, while maintaining linear scaling in syndrome size and strong parameter efficiency. The method outperforms MWPM and prior neural baseline QECCT in both accuracy and scalability. Similar gains hold on rotated surface codes, indicating generalization across codes.

**Strengths:**

1. The method is well-motivated and seems technically sound, combining learned decoding with constraint-preserving post-processing is a clear and sensible idea.

2. The results shown are strong: the decoder achieves near-ML thresholds on toric and rotated surface codes while maintaining linear scaling and good parameter efficiency.

3. The approach seems to generalize well across different noise models and code families.

**Weaknesses:**

I do not have sufficient background in quantum error correction to fully assess the novelty of the proposed method relative to prior decoders.
While the approach appears reasonable and the empirical performance is impressive, I am unable to evaluate the theoretical aspects of stabilizer enforcement or the loss formulation.

**Questions:**

-

---

### Author Response · Authors · 2025-11-25
**Author Response:  Revised SAQ-Decoder Paper**

We sincerely thank the reviewer for their thoughtful and insightful comments. We have uploaded the revised manuscript and believe the extensive revisions undertaken based on their critique have significantly improved the clarity, rigor, and context of the paper. We are confident the revisited version of the SAQ-Decoder paper addresses all concerns raised.

Below, we detail the major additions and clarifications made to the manuscript:
- **Architectural Clarity and Acknowledgment**:
  - Explicit clarification of the final output (section 4.3).
  - We have clarified key architectural aspects and contextualized our contribution by citing and elaborating on the differences (e.g., with AlphaQubit, CrossMPT, and Kai et al.), thereby ensuring that our paper's novelty is clear (sections 2 and 4).
- **Expanded Baselines and Generalization**:
  - Added comparison against the classical decoder BP+OSD-2, which SAQ-Decoder consistently outperforms across all tested conditions (Section 5.1).
  - Included the *SAQ-Decoder (No CPND)* variant to isolate architectural gain, demonstrating the raw dual-stream design already beats all baselines (Section 5.1).
  - Demonstrated framework generalization with new performance results for the Color Code and Repetition Code under circuit noise (Section 5.1).
  - Extended the rotational surface code evaluation to $L_{code}=11$ to provide a stronger demonstration of scalability toward larger code distances (Section 5.1).
- **Quantitative Metrics and Context**:
   - Added a condensed table (Section 5.2) and a detailed appendix (Appendix F) on FLOPs, Inference Time, and Parameters, demonstrating SAQ-Decoder's significant advantage in numerical efficiency and scalability compared to QECCT.
   - Added a comparative table confirming our $\mathbf{18.6\%}$ threshold is the highest reported among recent SOTA neural and classical decoders (Appendix E).
   - Resolved the confusion by explicitly defining all labels used in the Global Token Ablation figure (Section 5.2).

We believe these comprehensive revisions make the paper significantly stronger, unequivocally demonstrating that the SAQ-Decoder simultaneously achieves near-optimal accuracy and computational efficiency.

---

### Author Response · Authors · 2025-12-01
**Summary of Rebuttal and New Experiments for the Area Chair (part 1)**

We sincerely appreciate your diligence in navigating the unique and complex circumstances of this review cycle. Understanding the challenges involved, we used the rebuttal period to conduct extensive new experiments specifically targeting the empirical questions raised by the reviewers. We believe these additional results and revisions effectively address the initial concerns, serving to further validate the robustness and contribution of our work.

Below, we detail the major additions and clarifications made to the manuscript to address all reviewer concerns regarding baselines, scalability, and the precision of our claims (Please note that all Section, Appendix, Figure, and Table references cited below correspond to the revised manuscript.):

1. **Expanded Baselines Comparison**
   - **Threshold Dominance:** We added a comparative table (Appendix E, Table 3) confirming our **18.6%** threshold is the highest reported among recent scalable decoders, outperforming QECCT (17.8%), Astra ($\sim$17.0%), and SU-NetQD (16.3%):
     | Decoder / Paradigm               | Reported Threshold (Depolarizing Noise) $\uparrow$ |
|----------------------------------|-----------------------------------------|
| SAQ-Decoder (Our Work)             | **18.6%**                               |
| QECCT (Transformer Baseline) [1]   | 17.8%                                   |
| Astra (2024)                 [2]   | ~17%                                    |
| SU-NetQD (2025)              [3]   | 16.3%                                   |
| ML+UF (2022)                 [4]   | 16.2%                                   |
| MWPM (Classical Baseline)    [5]   | 16%                                     |
| BP-OSD-2 (Classical Baseline)  [6]   | ~16%                                    |
| UIUF (2024)                  [7]   | 15.6%                                   |

        This broader comparison confirms that SAQ-Decoder's performance is state-of-the-art.
   - **Neural Baselines:** We added a direct LER comparison **against SU-NetQD** (Appendix E, table 4). For Toric codes ($L_{code}=7, p=0.09$), SAQ achieves a **~29% reduction** in Logical Error Rate compared to SU-NetQD:
    | $L_{code}$   | p    | SAQ   | SU-NetQD | BP-OSD-2 | MWPM  |
    |-----|------|-------|----------|----------|-------|
    | 5   | 0.09 | **0.051** | 0.057    | 0.107    | 0.108 |
    | 5   | 0.13 | **0.177** | 0.191    | 0.280    | 0.279 |
    | 5   | 0.20 | **0.519** | 0.534    | 0.623    | 0.624 |
    | 7   | 0.09 | **0.019** | 0.028    | 0.072    | 0.069 |
    | 7   | 0.13 | **0.139** | 0.149    | 0.262    | 0.253 |
    | 7   | 0.20 | **0.548** | 0.581    | 0.680    | 0.676 |

    - **Classical Baselines:** We added comparisons **against BP-OSD-2**, which SAQ-Decoder consistently outperforms across all tested conditions (Section 5.1).

---

2. **Empirical Generalization and Scalability:**
    - **Extended Scalability:** We successfully trained and evaluated the decoder on Rotated Surface Code $L_{code}=11$ (Section 5.1, Figure 4), providing concrete proof of performance at larger, practically relevant distances.
    - **Code and Noise Types Generalization:** We demonstrated framework generalization with new performance results for **Color Codes**   ($L_{code}=3,5$) and **Repetition Codes** under circuit-level noise (Section 5.1). This new evidence confirms that SAQ-Decoder is a truly generalizable framework applicable to diverse stabilizer code. In particular, for the Color Code (L=5) at $p=2.00e-02$, our SAQ-Decoder is 17.0% better than QECCT and 64.2% better than MWPM.

---

> ### Author Response · Authors · 2025-12-01
> **Summary of Rebuttal and New Experiments for the Area Chair (part 2)**
>
> ---
>
> 3. **Quantitative Metrics and Context:**
>     -  **Efficiency Analysis:** We added a condensed table (Section 5.2, Table 1) and a detailed appendix (Appendix F) on FLOPs, Inference Time, and Parameters. We demonstrate that for $L_{code}=10$ (depolarizing), SAQ requires **3.5x fewer** parameters and **~5x fewer** FLOPs than the transformer baseline (QECCT). The following table presents a partial subset of our results, illustrating the computational advantages of SAQ over QECCT:
>         | Code Type   | $L_{code}$ | Noise | Decoder | FLOPs [M $\downarrow$] | Params [M $\downarrow$] | Time [sec $\downarrow$] |
>         | :---------- | :-: | :---: | :------ | ----------------------: | -----------------------: | -----------------------: |
>         | **Toric**   | 8   | ind   | SAQ     | **93.92**               | **1.22**                 | **5.01e-05**             |
>         |             |     |       | QECCT   | 228.45                  | 1.75                     | 2.73e-04                 |
>         |             |     | dep   | SAQ     | **259.15**              | **1.70**                 | **2.15e-04**             |
>         |             |     |       | QECCT   | 458.01                  | 3.42                     | 9.72e-04                 |
>         | **Toric**   | 10  | ind   | SAQ     | **143.76**              | **1.26**                 | **9.72e-05**             |
>         |             |     |       | QECCT   | 357.44                  | 2.55                     | 6.28e-04                 |
>         |             |     | dep   | SAQ     | **392.09**              | **1.85**                 | **4.50e-04**             |
>         |             |     |       | QECCT   | 717.61                  | 6.64                     | 2.33e-03                 |
>         | **Rot. Surf.** | 9 | ind  | SAQ     | **58.72**               | **1.20**                 | **2.35e-05**             |
>         |             |     |       | QECCT   | 143.84                  | 1.41                     | 1.23e-04                 |
>         |             |     | dep   | SAQ     | **154.71**              | **1.63**                 | **9.15e-05**             |
>         |             |     |       | QECCT   | 288.13                  | 2.08                     | 4.16e-04                 |
>         | **Rot. Surf.** | 11 | ind | SAQ     | **86.40**               | **1.22**                 | **4.49e-05**             |
>         |             |     |       | QECCT   | 215.33                  | 1.69                     | 2.51e-04                 |
>         |             |     | dep   | SAQ     | **228.54**              | **1.68**                 | **1.73e-04**             |
>         |             |     |       | QECCT   | 431.66                  | 3.18                     | 8.87e-04                 |
>
>     - **Architectural Gain Isolation:** We included the **SAQ-Decoder (No CPND)** variant to isolate architectural gain, demonstrating the raw dual-stream design already outperforms classical and neural baselines (Section 5.1). In particular, for the $L_{code}=10$ toric code under depolarizing noise at $p = 0.2$, taking SAQ (LER $= 6.06 \times 10^{-1}$) as a reference, MWPM and QECCT incur LER increases of approximately $26\%$ and $19\%$, respectively, whereas `SAQ (No CPND)` increases the LER by only $2.8\%$. This indicates that `SAQ (No CPND)` suffers only a very small marginal degradation relative to SAQ, while maintaining the high marginal gap from the classical and neural baselines.
>     - We resolved the confusion by explicitly defining all labels used in the Global Token Ablation figure (Section 5.2).
>
> ---
>
> 4. **Architectural Clarity and Acknowledgment**:
>     - Explicit clarification of the final output (Section 4.3).
>     - We have clarified key architectural aspects and contextualized our contribution by citing and elaborating on the differences (e.g., with AlphaQubit, CrossMPT, and Kai et al.), thereby ensuring that our paper's novelty is clear (Sections 2 and 4).

---

> > ### Author Response · Authors · 2025-12-01
> > **Summary of Rebuttal and New Experiments for the Area Chair (part 3)**
> >
> > ---
> >
> > 5. **Refinement of Claims (Addressing Over-Claiming Concerns):**
> >     - Scoped Applicability: We have revised the Section 5 introduction to claim *"adaptability to distinct lattice geometries"* rather than universal applicability. We support this with new empirical validation on Color Codes and Repetition Codes, verifying the framework's robustness across diverse topological structures.
> >     - Precision on Scalability: We refined the abstract to specify that our decoder exhibits *"linear computational scalability with respect to the syndrome size"*. This acknowledges the geometric scaling of surface codes while correctly identifying our architectural advantage over super-linear classical matchers.
> >     - Precision on Performance: We refined the claim of *"near-optimal"* performance to the quantitative statement that our method *"closely approaches theoretical Maximum Likelihood (ML) bounds"* (e.g., 18.6% vs. 18.9%).
> >
> > We sincerely thank the Area Chair and reviewers for their time and constructive feedback, which has strengthened this work. The rigorous review process prompted us to refine the precision of our claims and expand our empirical validation, resulting in a much more robust manuscript.
> >
> > ## References
> >
> > 1. Choukroun & Wolf, *“Deep quantum error correction”*, AAAI (2024).
> >
> > 2. Arshpreet Singh Maan and Alexandru Paler, *“Machine Learning Message-Passing for the Scalable Decoding of QLDPC Codes”* (2024).
> >
> > 3. Zhang, Wei-Wei and Xia, Zhuo and Zhao, Wei and Pan, Wei and Shi, Haobin, *“Self-attention U-Net decoder for toric codes”*, Physical Review Applied (2025).
> >
> > 4. Meinerz, Kai and Park, Chae-Yeun and Trebst, Simon, *“Scalable Neural Decoder for Topological Surface Codes”*, Physical Review Letters (2022).
> >
> > 5. D. S. Wang and A. G. Fowler and A. M. Stephens and L. C. L. Hollenberg, *“Threshold error rates for the toric and surface codes”* (2009).
> >
> > 6. `ldpc` package (BP-OSD Decoder), our findings.
> >
> > 7. Lin & Lai, *“Union-Intersection Union-Find for Decoding Depolarizing Errors in Topological Codes”*, IEEE Journal on Selected Areas in Information Theory (2025).

---

### Meta-Review · Area_Chair_aXrg · 2026-01-10

**Summary:**

I’ve read all the review comments and the authors’ detailed responses to those review comments. Because of the specialized technicality of this paper, one of the reviewers was not able to provide detailed review comments (and another one the system invited was also not able to provide reviews).

But we do have two excellent reviewers who provided very clear and insightful comments about the submission. I believe, by answering and addressing those comments, the authors were able to improve their paper’s representation significantly.

The only remaining issues are the following:
(1) the scalability for code distances up to 10 (and the revision added the distance of L=11, while L>12 is not available);
(2) the scalability of the overall algorithm (the reviewer pointed out the fundamental limits on achieving the quadratic complexity) – while the authors clarified their linear scalability is limited to the decoder part only; and
(3) the overall significance of this work’s contribution to the field – though the review comments are all addressed (and a lot of review comments were valid and correct), the undecided part is, with these modifications to the work, whether the work as is remains significant enough. Clearly one of the expert reviewers didn’t think so, while the other expert reviewer didn’t confirm.

So the two expert reviewers’ average score remains to be at 4: marginally below the acceptance threshold), which is probably justified.

However, I also found the authors efforts to respond to the reviewers’ comments to be genuine and their answers and modification of the submissions are mostly to the point. It seemed reasonable to say that this submission with the modification would still contribute new knowledge to the field.

Therefore, I would like to suggest the increase the score of the rating and recommend it as “accept for poster”.

**Reviewer Concerns:**

Reviewers had concerns about the clarity of the wring, the over-claims of the contributions, comparison with state-of-the-arts, acknowledgement of others similar work and the differences between the submission and others work, and the experiment settings. The authors have addressed most of them well.
The only remaining issues are the following:
(1) the scalability for code distances up to 10 (and the revision added the distance of L=11, while L>12 is not available);
(2) the scalability of the overall algorithm (the reviewer pointed out the fundamental limits on achieving the quadratic complexity) – while the authors clarified their linear scalability is limited to the decoder part only; and
(3) the overall significance of this work’s contribution to the field – though the review comments are all addressed (and a lot of review comments were valid and correct), the undecided part is, with these modifications to the work, whether the work as is remains significant enough. Clearly one of the expert reviewers didn’t think so, while the other expert reviewer didn’t confirm.

So the two expert reviewers’ average score remains to be at 4: marginally below the acceptance threshold), which is probably justified.

**Reviewer Scores:**

Both reviewers will probably maintain their respective scores

---

### Decision · Program_Chairs · 2026-01-26

Accept (Poster)